# Easy-to-Hard Generalization:
# Scalable Alignment Beyond Human Supervision

**Zhiqing Sun**[1][*], **Longhui Yu**[2][*], **Yikang Shen**[3], **Weiyang Liu**[4,5],
**Yiming Yang**[1][†], **Sean Welleck**[1][†], **Chuang Gan**[3,6][†]

[1]Carnegie Mellon University, [2]Peking University, [3]MIT-IBM Watson AI Lab
[4]University of Cambridge, [5]Max Planck Institute for Intelligent Systems, [6]UMass Amherst
Code: Edward-Sun/easy-to-hard

## Abstract

Current AI alignment methodologies rely on human-provided demonstrations or judgments, and the learned capabilities of AI systems would be upper-bounded by human capabilities as a result. This raises a challenging research question: How can we keep improving the systems when their capabilities have surpassed the levels of humans? This paper answers this question in the context of tackling hard reasoning tasks (*e.g.*, level 4-5 MATH problems) via learning from human annotations on easier tasks (*e.g.*, level 1-3 MATH problems), which we term as *easy-to-hard generalization*. Our key insight is that an evaluator (reward model) trained on supervisions for easier tasks can be effectively used for scoring candidate solutions of harder tasks and hence facilitating easy-to-hard generalization over different levels of tasks. Based on this insight, we propose a novel approach to scalable alignment, which firstly trains the (process-supervised) reward models on easy problems (e.g., level 1-3), and then uses them to evaluate the performance of policy models on hard problems. We show that such *easy-to-hard generalization from evaluators* can enable *easy-to-hard generalizations in generators* either through re-ranking or reinforcement learning (RL). Notably, our process-supervised 7b RL model and 34b model (reranking@1024) achieves an accuracy of 34.0% and 52.5% on MATH500, respectively, despite only using human supervision on easy problems. Our approach suggests a promising path toward AI systems that advance beyond the frontier of human supervision.

## 1 Introduction

Rapid advancements in large language models (LLMs) indicate that in the near future, highly sophisticated AI systems could surpass human capabilities in certain areas, significantly enhancing our capabilities in solving harder problems beyond the levels we can currently solve [47, 49]. Since the current AI alignment methods mostly rely on either supervised fine-tuning (SFT) with human-provided demonstrations [59, 78, 14] or reinforcement learning from human feedback (RLHF) [97, 68, 50], their capabilities would be inherently limited as humans cannot always provide helpful demonstrations or supervision on the hard tasks beyond their expertise [64].

In order to build future AI systems for tackling complex challenges, such as advancing scientific knowledge, it is crucial to develop new approaches for *scalable oversight* challenge, i.e., to supervise the AI systems that can potentially outperform humans in most skills [9]. The key question is:

- *Can we limit human supervision to easier tasks, yet enable the model to excel in harder tasks?*

---

[*]Equal contributions as leading authors.
[†]Equal contributions as senior authors.

38th Conference on Neural Information Processing Systems (NeurIPS 2024).

**Traditional Alignment**

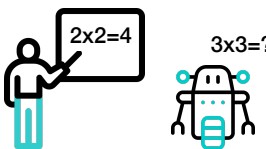

humans supervise strong models on hard tasks

**Scalable Alignment (Superalignment)**

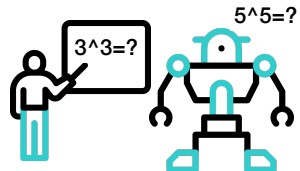

humans cannot reliably supervise **superhuman** models on the **hardest** tasks

**Burns' Analogy on Weak-to-Strong Generalization**

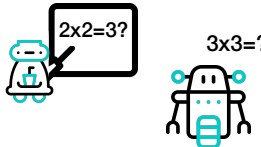

**weak** models unreliably supervise **strong** models on hard tasks that humans can evaluate

**Our Analogy on Easy-to-Hard Generalization**

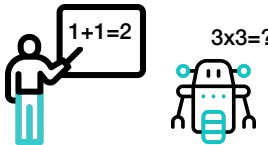

humans reliably supervise strong models on **easy** tasks and evaluate them on **hard** tasks

Figure 1: Illustration of different alignment scenarios: **traditional alignment** relies on human demonstrations or judgements [50]; **scalable alignment** [9] assumes that humans cannot reliably supervise smarter-than-human models; **weak-to-strong generalization** [11] focuses on using weak models with unreliable labels to supervise strong models; Our proposed **easier-to-general generalization** focuses on the transfer of rewarding policies from weak models to harder tasks.

We refer to this scenario as *Easy-to-Hard Generalization* [63, 95, 11, 29]. This setting requires no human supervision on the harder tasks, which differs from existing work that either enhances humans' ability to verify the outputs of AI systems [81, 60, 9, 57] or enables weak-to-strong generalization via a teacher that only offers unreliable or noisy supervision [11].

The most basic form of easy-to-hard generalization can be achieved by training the policy models (i.e., generator) using supervised fine-tuning (SFT) or in-context learning (ICL) on easy tasks [55, 10], and expect this will unlock the ability to perform well on hard tasks. However, it has been observed that SFT or ICL training of generators on easy tasks often fails to generalize to hard tasks [71, 24, 95]. We hypothesize and show that methods beyond these can enable stronger degrees of easy-to-hard generalization. Our intuition is guided by the observation that *evaluation is easier than generation* [34, 46], so an evaluator may offer a degree of easy-to-hard generalization that is useful for improving a generator. If that is true, we can first train a verifier on easy tasks, then make use of its generalization ability to supervise the generator on hard tasks.

Complex tasks can often be broken down into smaller steps [95] and verified by validating the individual steps – a strategy that is commonly employed in solving mathematical problems [74, 40, 73]. Inspired by this, we train outcome-supervised and process-supervised reward models [74, 85, 75, 40] as our easy-to-hard evaluators. The training dataset is often comprised of a set of labeled easy tasks, each with a question and a high-quality solution[1], paired with a set of unlabeled hard tasks that are represented only by their questions. This simulates the practical setting of having numerous problems with known solutions, as well as significant unresolved challenges, such as the Millennium Prize Problems [12], which present challenging open problems. The pivotal aspect of easy-to-hard generalization thus lies in how we effectively leverage the capabilities of easier-level models in solving harder problems.

Our investigation includes to training policy and reward models on the easy (i.e., level 1-3) portion of the PRM800K [40] dataset, and comparing the performance of majority voting with the policy model

---

[1]We assume that human supervision is of high quality on the easy tasks in general.

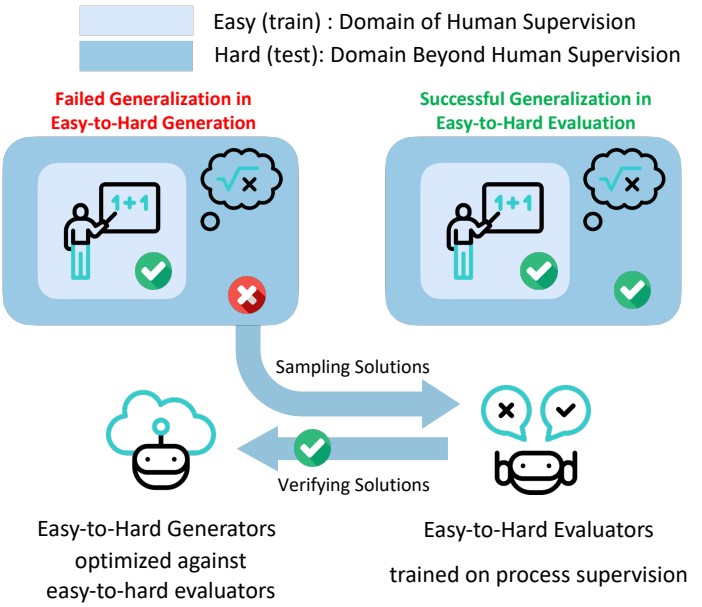

Figure 2: We first train the evaluator with process supervision or outcome supervision (which simulates the process supervision) to enable easy-to-hard evaluation, and then use it to facilitate easy-to-hard generation via re-ranking or RL.

only and weighted majority voting with the policy model and PRMs (Process-supervised Reward Models). We also introduce the *Outcome & Process Reward Model (OPRM)*, which harnesses the complementary strengths of outcome reward models (ORMs) and process reward models (PRMs): judging if each step in reasoning is correct (like PRMs do) and deciding if the final answer is right (like ORMs do). Our findings reveal a marked performance improvement with the inclusion of reward models, especially on the hard (i.e., level 4-5) portion of the MATH500 test set. This improvement indicates that easier-level evaluators can maintain their effectiveness on harder tasks. We have similar observations in our experiments on the MetaMath dataset [86] and the Math-Shepherd dataset [75].

We further investigate the use of the easy-to-hard evaluator as a reward model in reinforcement learning, where the evaluator provides targeted, step-by-step guidance in solving hard problems. We have an intriguing finding that *training with human supervision only on the easy tasks (i.e., training with Level 1-3 problems and answers) can outperform both SFT and Final-Answer RL training on the full dataset (Level 1-5)*. This finding underscores the potential of using easy-to-hard evaluation to improve easy-to-hard generators, particularly when dealing with varied levels of task complexity.

## 2 Related Work

### 2.1 Scalable Oversight

While present-day models operate within the scope of human assessment, future, more advanced models may engage in tasks that are beyond human evaluation capabilities. This raises a concern that such models might prioritize objectives other than maintaining accuracy (Andreas [3], Perez et al. [53], Sharma et al. [64], Wei et al. [80]). To address this, a branch of research develops techniques to enhance the human capacity to supervise such models, such as via using AI to evaluate the work of other AIs [1, 38, 60, 9]. Our setting differs from enhancing human oversight; instead, we focus on enabling models to excel in hard tasks where human supervision may not be available. This also differs from weak-to-strong generalization [11], where human supervision may be available, but not reliable, on hard tasks. However, our framework aligns with the "sandwiching" concept proposed for measuring progress in scalable oversight, which involves domain experts evaluating the outputs of AI-assisted non-experts [18, 9, 57].

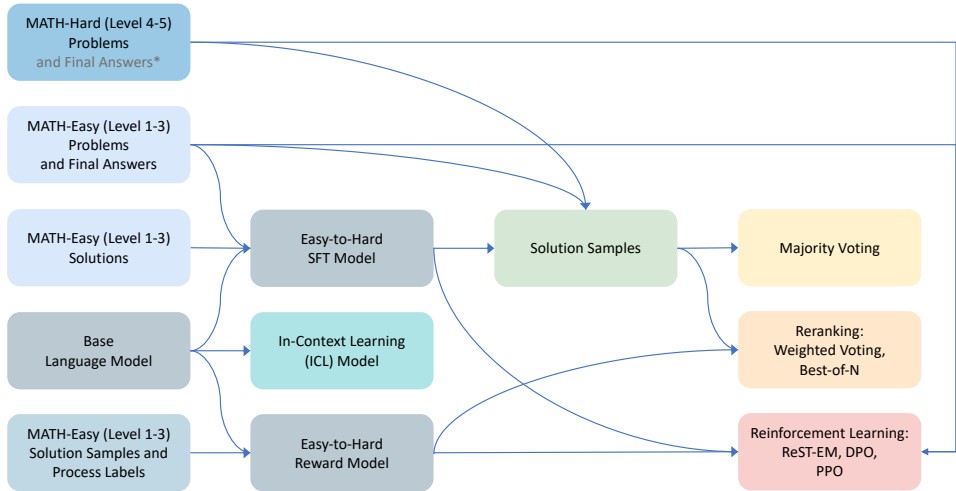

Figure 3: The overview diagram of our methods: the different components of modeling and training and how they are interconnected.

## 2.2 Compositional Generalization

Compositional generalization is a fundamental aspect of how language works [13]. It refers to the ability to understand and utilize novel combinations based on the understanding of basic concepts and a limited number of their combinations [23]. Recently, least-to-most prompting [95, 20] teaches language models how to solve a complex problem by reducing it to a series of easier sub-problems, achieving easy-to-hard generalization on semantic parsing tasks like SCAN [37] and CFQ [35] with perfect generalization accuracy. In addition, least-to-most prompting has also been successful in mathematical reasoning tasks, specifically in datasets like GSM8K [16] and DROP [21], by teaching language models to solve problems more difficult than those seen in the prompts. This success not only underscores the capacity of language models to effectively break down complex tasks into simpler sub-tasks Perez et al. [51], but also demonstrates their generalization capability in solving these sub-problems.

## 2.3 Easy-to-Hard Generalization

Past work has evaluated easy-to-hard generalization by training easy-to-hard generators on easy tasks using supervised finetune-tuning (SFT) or in-context learning (ICL) [55, 10]. Nevertheless, Swayamdipta et al. [71] showed that the BERT model performs poorly on common-sense reasoning when only trained on easy data. Fu et al. [24] showed similar results for ICL on reasoning tasks like GSM8K [17]. In concurrent work, Hase et al. [29] evaluate the performance of easy-to-hard generators on more datasets and models, and find that ICL or SFT on easy tasks is a strong baseline for multiple-choice tasks like ARC [15] and MMLU [30]. In contrast, we evaluate the easy-to-hard generation performance on the more challenging MATH dataset [32], and show that easy-to-hard evaluation can improve a generator's easy-to-hard generalization beyond ICL and SFT. Iterative machine teaching [43] gives theoretical justification to show that training classifiers from easy to hard examples yield better generalization.

## 3 Methodology

We study the easy-to-hard generalization problem: how can we enable capabilities beyond human supervision? Specifically, we explore the efficacy and scalability of various easy-to-hard methodologies on competition-level mathematical problem-solving problems (MATH; Hendrycks et al. 32). This dataset is suitable for our study since it explicitly categorizes problems across five difficulty levels. We consider levels 1-3 as "easy" tasks, encompassing both the problems and their respective solution demonstrations, along with the correct answers. Conversely, levels 4-5, characterized by their more complex nature, are treated as "hard" tasks and are represented solely by their questions. The MATH

dataset's difficulty distribution roughly follows a $1 : 2 : 2 : 3 : 3$ ratio across levels 1 to 5. So our division maintains a balanced number of easy and hard tasks.

The remainder of the paper aims to answer following research questions:

**RQ1:** How do generators generalize from easy to hard?

**RQ2:** How do evaluators generalize from easy to hard?

**RQ3:** If evaluators generalize better than generators, how can we take advantage of this to enable stronger easy-to-hard generalization in generators?

## 3.1 Setup

**Dataset** MATH [32] is a dataset of 12,500 challenging competition mathematics problems, where 7,500 of them are training problems and 5,000 are originally used for testing. Following Lightman et al. [40], Wang et al. [75], we use the identical subset of 500 representative problems (i.e., MATH500) as our test set, uniformly sample another 500 problems for validation, across all five difficulty levels, and leave the rest 4,000 MATH test split problems combined with the original 7,500 MATH training split problems as our training set.

**Simulated Human Demonstrations** While the original MATH dataset provides full step-by-step solutions, these solutions typically skip many chain-of-thought steps [79], which can be hard for language models to directly imitate[2]. Instead, we consider filtered PRM800K [40] and MetaMATH [86] as our SFT training data: the former is generated by a Minerva-style base GPT-4 model using few-shot prompting after filtering the correct answers [39, 48], while the latter is generated by ChatGPT [47]. We keep all the GSM8K data in the MetaMATH dataset since they are typically easier than the problems in MATH. PRM800K comes with human annotated process labels, while for MetaMath, we use Math-Shepherd as the corresponding process labels [75].

## 3.2 Generators

For a given dataset (e.g., a variant of MATH), we consider the following generator models:

**Full & Hard ICL** Full in-context learning (ICL) is a base model prompted with exemplars sampled from all difficulty levels, or only from the level 5 [24].

**Easy-to-Hard ICL** This model is prompted with exemplars from easy problems. This baseline evaluates the degree to which a model can solve problems more difficult than those seen in the prompts [95].

**Full SFT** As prior work suggests that finetuning should outperform prompting alone [68, 52, 50], the full supervised fine-tuning (SFT) model is typically considered as a ceiling that a model can achieve on a type of task [11, 29].

**Easy-to-Hard SFT** This generator model is trained only on the easy tasks. Prior work suggests that it can generalize to hard tasks but with some degeneration in performance [71].

The generator models are evaluated in greedy decoding and self-consistency (also known as majority voting) settings [76].

## 3.3 Evaluators

Similarly, we consider the following evaluator models that can be trained either on the easy tasks only, or on the full dataset. Notably, unlike final-answer rewards, reward models trained on easy tasks can be applied to evaluate solutions to hard problems.

**Final-Answer Reward** is a symbolic reward that provides a binary reward based on the accuracy of the model's final answer. The matching is performed after normalization[3].

---

[2]Hendrycks et al. [32] found that having models generate MATH-style step-by-step solutions before producing an answer actually decreased accuracy.

[3]https://github.com/openai/prm800k/blob/main/prm800k/grading/grader.py

Table 1: Easy-to-hard generalization of generators. We compare generator performance under various decoding settings. PRM800K and METAMATH indicate the SFT training data and ICL exemplars. Evaluations are performed on the same MATH500 test set.

| | | PRM800K | | | METAMATH | | |
| | | GREEDY | MAJ@16 | MAJ@256 | GREEDY | MAJ@16 | MAJ@256 |
|---|---|---|---|---|---|---|---|
| LLEMMA-7B | FULL ICL | 12.8 | 15.6 | 20.8 | 16.4 | 18.4 | 25.6 |
| | HARD ICL | 12.6 | 18.0 | 27.0 | 16.6 | 19.0 | 27.0 |
| | EASY-TO-HARD ICL | 14.0 | 17.6 | 24.4 | 14.2 | 17.4 | 26.8 |
| | FULL SFT | 20.6 | 32.0 | 36.2 | 31.4 | 40.2 | 41.6 |
| | EASY-TO-HARD SFT | 19.8 | 31.6 | 36.0 | 30.0 | 38.6 | 42.4 |
| LLEMMA-34B | FULL ICL | 18.6 | 23.6 | 36.0 | 20.6 | 28.8 | 39.2 |
| | HARD ICL | 15.8 | 21.4 | 34.2 | 21.8 | 26.4 | 38.6 |
| | EASY-TO-HARD ICL | 18.2 | 25.2 | 36.8 | 19.8 | 26.8 | 37.2 |
| | FULL SFT | 25.6 | 41.8 | 46.4 | 35.4 | 44.2 | 45.6 |
| | EASY-TO-HARD SFT | 24.8 | 40.8 | 46.0 | 32.2 | 42.6 | 43.4 |

**Outcome Reward Model (ORM)** is trained on the Final-Answer rewards. Following Cobbe et al. [16], Uesato et al. [74], Lightman et al. [40], we train the reward head to predict on every token whether the solution is correct, in a similar sense to a value model [85]. At inference time, we use the ORM's prediction at the final token as the reward of the solution.

**Process Reward Model (PRM)** is trained to predict whether each step (delimited by newlines) in the chain-of-thought reasoning path is correct. The labels are usually labeled by humans [74, 40] or estimated with rollouts [65, 75].

**Outcome & Process Reward Model (OPRM)** Building on the distinct advantages of ORMs and PRMs, we introduce the *Outcome & Process Reward Model (OPRM)*, which harnesses the complementary strengths of both. OPRM is trained on the mixed data of ORMs and PRMs. Specifically, it evaluates the correctness of each intermediate reasoning step, akin to PRMs, while also assesses the overall solution's accuracy at the final answer stage, mirroring the functionality of ORMs.

### 3.4 Optimizing Generators Against Evaluators

Finally, given a generator model (i.e., policy model) and a evaluator model (i.e., reward model; RM), we optimize the generator against the evaluator using either re-ranking or reinforcement learning.

**Best-of-$n$ (BoN)**, also known as rejection sampling, is a reranking approach that sample multiple solutions from the generator and selects one with the highest RM score.

**Weighted Voting** is similar to majority voting or self-consistency [76], but weights each solution according to its RM score [74].

**Reinforcement Learning (RL)** We consider three online/offline RL variants, Reinforced Self-Training (ReST) [28, 67], Direct Policy Optimization (DPO) [56], and Proximal Policy Optimization (PPO) [62]. Due to the space limit, please find their detailed description in Appendix B.

### 3.5 Evaluation Metrics

In this study, we have chosen not to establish terms analogous to the weak-to-strong performance gap recovery (PGR) as discussed in Burns et al. [11] or the easy-to-hard supervision gap recovery (SGR) highlighted by Hase et al. [29]. This decision is based on our observations that sometimes, models trained exclusively on simpler tasks—particularly when employing RL training—can outperform those trained across the entire spectrum of problem difficulties. Therefore, we mainly focus on the absolute and relative performance of generators (optionally optimized by the evaluator) on the MATH500 test set [40].

### 3.6 Implementation Details

**Base Language Model** Llemma is a large language model for mathematics [6], which is continue pretrained from Code Llama [58] / LlaMA-2 [72]. We use both 7b and 34b variants in our experiments.

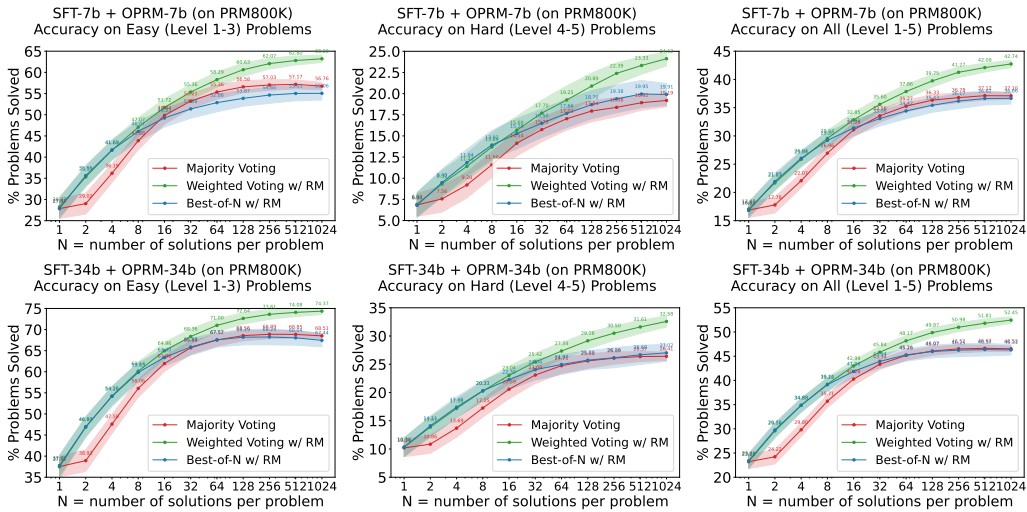

Figure 4: Easy-to-hard generalization of 7b (upper) and 34b (lower) evaluators. Both SFTs and RMs are trained on the easy data. We found that PRMs trained on easy tasks can significantly improve the re-ranking (i.e., weighted voting) performance on hard tasks. The shaded margin of the curve plot in this paper represents the performance variance.

**SFT / RL / Reward Model** We fine-tune all models in full fine-tuning with frozen input-output embedding layers and normalization layers. RMs are initialized from the base model, and have an added scalar head to output the reward. In PPO training, we initialize the value model from the reward model.

**Hyper-parameters** Due to the space limit, our training hyper-parameters can be found in Appendix. C.

## 4 Main Results

### 4.1 Easy-to-Hard Generalization of Generators

In Table 1, we compare the easy-to-hard generalization performance of the generators under various decoding settings:

**Supervised Fine-Tuning (SFT) outperforms In-Context Learning (ICL):** This is consistent with prior work [68, 50, 74]. We also find that the performance of ICL has larger variance than SFT with respect to data ordering (or random seeds) [19, 93].

**SFT data quality impacts easy-to-hard generalization:** PRM800K data is generated by a base (unaligned) GPT-4 model through few-shot prompting and is thus of lower quality than well-aligned ChatGPT-generated MetaMATH data. We find that only MetaMath-trained models have certain easy-to-hard gaps (e.g., 16.6 v.s. 14.2 in MetaMath-7b-ICL), while such gaps in PRM800K-trained models are very small (less than 1%), or even inverted in the ICL setting. We hypothesize that low-quality SFT data may only teach the model the format of the task [59, 78, 76], while high-quality (imitation) SFT data can teach the model the principles of solving the task [70, 27]. Nevertheless, the strongest performance is achieved by full SFT on the high-quality MetaMath data (35.4), showing an unignorable difference, with a gap of up to 3.2, compared to its easy-to-hard SFT counterpart (32.2).

### 4.2 Easy-to-Hard Generalization of Evaluators

The primary metric we use to assess the effectiveness of our process reward model is not the average accuracy of verifying each step in a solution but rather the overall performance achieved through re-ranking methods (See discussion in Sec. 3.5). We first use re-ranking to evaluate the easy-to-hard generalization performance of evaluators.

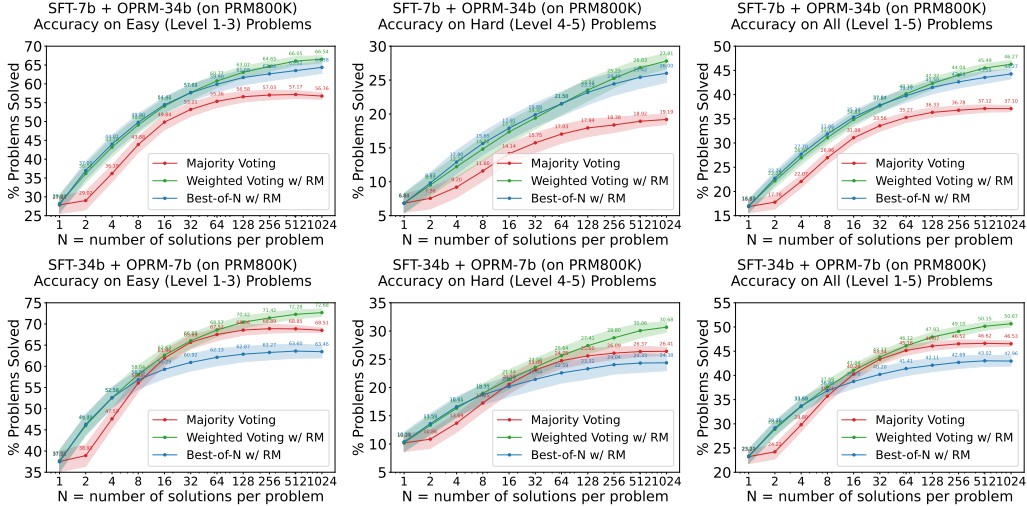

Figure 5: Easy-to-hard generalization of evaluators applied to generators of different sizes. We evaluated 7b generator + 34b evaluator (upper) and 34b generator + 7b evaluator (lower). Both SFTs and RMs are trained on the easy data.

### 4.2.1 Re-ranking

We consider two re-ranking strategies: Best-of-$n$ (or rejection sampling) and Weighted Voting. In our easy-to-hard generalization setting, both SFT models and Reward Models (RMs) are trained on easier tasks (levels 1-3), but evaluated on all difficulty levels (1-5). We compare the performance between majority voting (SFT only) and re-ranking (SFT + OPRM) on the PRM800K dataset in Figure 4-5, and the performance of different reward models (PRMs, ORMs, & OPRMs) on the PRM800K dataset in Figure 8-9. Specifically, we use `min` as the reward aggregation function for best-of-$n$ and `prod` for weighted voting[4]. The figures illustrate the performance of different decoding strategies or reward models under the same number of sampled solutions per problem. We have the following findings:

**OPRMs outperforms ORMs and PRMs**   This confirms our hypothesis that Process Reward Models (PRMs) and Outcome Reward Models (ORMs) capture different aspects of task-solving processes. By integrating the strengths of both PRMs and ORMs, Outcome & Process Reward Models (OPRMs) demonstrate superior performance. However, follow-up experiments conducted on the MetaMath/Math-Shepherd datasets do not demonstrate significant improvements from incorporating additional ORM training examples. This lack of enhancement may be attributed to the fact that Math-Shepherd is already generated from final-answer rewards. This suggests that there remains a substantial difference between process rewards labeled by humans (e.g., PRM800K) and those generated automatically (e.g., Math-Shepherd).

**Weighted voting outshines Best-of-$n$**   This finding diverges from past research where minimal performance differences were observed between weighted voting and Best-of-$n$ [40, 74]. Our hypothesis is that this discrepancy arises from our specific experiment, which involves training a less powerful base model (Llemma; Azerbayev et al. 6) on more difficult tasks (MATH; Hendrycks et al. 32). This setup might diminish the effectiveness of the reward model, potentially leading to an over-optimization of rewards [25]. Given these insights, weighted voting is preferred as the primary re-ranking method for further discussions. Nevertheless, Best-of-$n$ still achieves competitive performance to majority voting when producing only one full solution. In Figure 5, we also find that the 34b evaluator can significantly improve the 7b generator, while the 7b evaluator can still improve the performance of the 34b generator.

**Greater effectiveness of re-ranking on harder tasks:**   Weighted voting not only consistently surpasses majority voting but also shows a more pronounced advantage on harder tasks. This

---

[4]See more detailed analysis of reward aggregation functions in Appendix. L.

Table 2: Comparing reinforcement learning (RL) approaches for easy-to-hard generalization. All methods are of 7b size and evaluated with greedy decoding.

| | RL Data | Reward | | Accuracy | | |
| | | Final-Answer | Process RM | Easy (level 1-3) | Hard (level 4-5) | All |
|---|---|---|---|---|---|---|
| *(SFT / PRM trained on level 1-3 of PRM800K)* | | | | | | |
| SFT | | | | 28.2 | 12.2 | 19.8 |
| ReST-EM | Easy | Easy | × | 33.2 | 12.6 | 22.4 |
| Iterative DPO | Easy | Easy | √ | 42.0 | 12.2 | 26.4 |
| PPO | Easy | Easy | × | 42.0 | 14.1 | 27.4 |
| PPO | All | Easy | √ | **45.4** | **14.9** | **29.4** |
| *(SFT / PRM trained on level 1-5 of MetaMath / Math-Shepherd)* | | | | | | |
| Llemma-based SFT SoTA (Ours) | | | | 51.7 | 13.7 | 31.4 |
| Previous RL SoTA [75] | | | | - | - | 33.0 |
| *(SFT / PRM trained on level 1-3 of MetaMath / Math-Shepherd)* | | | | | | |
| SFT | | | | 44.1 | 14.9 | 28.8 |
| ReST-EM | Easy | Easy | × | 50.4 | 14.5 | 31.6 |
| Iterative DPO | Easy | Easy | √ | **53.8** | **16.0** | **34.0** |
| Iterative DPO | All | Easy | √ | 49.6 | 10.7 | 29.2 |
| PPO | Easy | Easy | × | 50.8 | 15.3 | 32.2 |
| PPO | All | Easy | √ | **53.8** | **16.0** | **34.0** |

observation leads to the conclusion that *evaluators demonstrate better easy-to-hard generalization capabilities in comparison to generators*. This motivates us to explore RL approaches that optimize the generator against the evaluator to further improve the performance of easy-to-hard generation.

### 4.2.2 Reinforcement Learning (RL)

Given the conclusion above, an important question arises: how can evaluators once again assist generators in achieving enhanced easy-to-hard generalization capabilities? We further investigate the enhancement of policy models through RL, utilizing easy-to-hard evaluators as reward models. Similar to re-ranking, SFT and PRM are only trained on easy data. For a fair comparison between PRM800K and MetaMath, we only use vanilla PRMs in the RL training. All the RL methods use the validation accuracy for selecting the best checkpoint[5]. Our comparison spans offline (ReST & DPO) and online (PPO) RL algorithms under two training conditions:

**Easy Questions & Easy Final Answers.** The SFT model samples from easy questions and receives the corresponding Final-Answer and optional PRM rewards.

**All Questions & Easy Final Answers.** This assumes access to a range of easy and hard problems for RL training, with rewards for hard tasks solely provided by the easy-to-hard evaluator.

Based on the results reported in Table 2, we have the following findings:

**DPO and PPO excel over ReST.** Among the RL algorithms trained on the PRM800K dataset, PPO emerges as the most effective, significantly surpassing both ReST and DPO. On the MetaMATH dataset, PPO and DPO achieve top performance, while ReST shows only marginal improvements over the SFT baseline. The comparative analysis between DPO and PPO across the PRM800K and MetaMATH datasets indicates that while DPO's efficacy is on par with PPO given a high-quality SFT model as initialization, PPO's effectiveness is less contingent on the quality of the underlying SFT model [50, 56].

**PRM rewards are more beneficial than Final-Answer rewards for hard tasks.** Notably, models trained with PRM rewards with human supervision on the easy tasks (achieving a top performance of 34.0) outperform the previous state-of-the-art model trained across all task levels (33.0). This highlights the effectiveness of leveraging easy-to-hard evaluations to improve generator performance across varying task difficulties.

---

[5]This includes stopping iterations in ReST-EM and iterative DPO, and stopping online steps in PPO.

Table 3: Easy-to-hard generalization of evaluators on coding problems (APPS). Both SFTs and RMs are trained on the easy (Introductory) data. We found that ORMs trained on easy tasks can improve the re-ranking (Best-of-N) performance on hard (Interview & Competition) coding problems.

| | SFT / ORM TRAIN DATA | DECODING | AVERAGE ACCURACY (%) | | | | STRICT ACCURACY (%) | | | |
|---|---|---|---|---|---|---|---|---|---|---|
| | | | INTRO. | INTER. | COMP. | ALL | INTRO. | INTER. | COMP. | ALL |
| CODE LLAMA - 7B | ALL | GREEDY | **31.4** | 15.5 | **12.2** | **18.0** | 17.0 | 2.3 | 2.0 | 5.2 |
| | EASY | GREEDY | 26.8 | 14.1 | 9.5 | 15.7 | 11.0 | 3.0 | 0.0 | 4.0 |
| | EASY | BEST-OF-1 | 25.4 | 12.0 | 0.1 | 13.5 | 16.0 | 2.7 | 0.0 | 4.8 |
| | EASY | BEST-OF-4 | 27.1 | 13.8 | 8.1 | 15.3 | 14.0 | 4.0 | 0.0 | 5.2 |
| | EASY | BEST-OF-16 | 29.7 | **16.3** | 11.3 | **18.0** | **19.0** | **5.0** | **3.0** | **7.4** |
| CODE LLAMA - 34B | ALL | GREEDY | 37.6 | 19.9 | **11.3** | 21.7 | 22.0 | 5.0 | 2.0 | 7.8 |
| | EASY | GREEDY | 33.9 | 19.4 | 8.5 | 20.1 | 21.0 | 6.0 | 1.0 | 8.0 |
| | EASY | BEST-OF-1 | 28.5 | 14.5 | 4.4 | 15.3 | 21.0 | 3.3 | 0.0 | 6.2 |
| | EASY | BEST-OF-4 | 36.3 | 21.3 | 10.5 | 22.1 | 24.0 | 8.7 | 1.0 | 10.2 |
| | EASY | BEST-OF-16 | **45.9** | **25.8** | 10.0 | **26.6** | **30.0** | **10.7** | **3.0** | **13.0** |

### 4.3 Easy-to-Hard Generalization on the Coding Domain

We conduct further experiments in the coding domain with the APPS dataset [31]. Similarly to Lightman et al. [40], we sub-sampled 500 questions from the original test set of APPS as our test set. Specifically, we sub-sampled 100 Introductory questions, 300 Interview questions, and 100 Competition questions, following the original distribution in the test set.

In Table 3, we compare the performance of SFT-trained Code Llama [58] (7b & 34b) with greedy decoding and best-of-N approach. In the latter, an Outcome Reward Model (ORM) of the same model size is trained to select the best coding one from N sampled solutions.

We found that while the reward model is only trained on the outcome supervision of easy (Introductory) data, it significantly improves the model performance on hard (Interview & Competition) data. These findings extend the premise of easy-to-hard generalization beyond the confines of mathematical reasoning, suggesting its applicability across diverse domains.

## 5 Conclusion

Our study advances the field of AI alignment by demonstrating the potential of easy-to-hard generalization, where models trained on simpler tasks can be guided to solve more complex problems without direct human supervision on these harder tasks. Through the use of (process-supervised) reward models for evaluating and enhancing policy models, we show that evaluators can facilitate this form of generalization, outperforming traditional training methods. Our findings highlight the effectiveness of re-ranking strategies and reinforcement learning (RL) in leveraging evaluators for performance gains on difficult tasks. This approach presents a promising direction for developing AI systems capable of surpassing human problem-solving capabilities, suggesting a scalable alignment method that could enable AI to independently advance knowledge in complex domains.

While our study provides valuable insights into easy-to-hard generalization and the potential of process-supervised reward models, there are limitations to consider. These include the focus on specific model sizes and datasets, the domain specificity of reasoning tasks, and the need for further research on the long-term implications and robustness of the method.

## 6 Acknowledgement

This work is supported by OpenAI Superalignment Fast Grants and Microsoft Accelerate Foundation Models Research (AFMR) Initiative. Additionally, ZS thanks Google PhD Fellowship; SW thanks NSF SCALE (NSF DMS 2134012) and Convergent Research.

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

# A Additional Related Work

## A.1 Rationale-Augmented (Mathematical) Reasoning

Ling et al. [41] pioneer the work of solving math word problems by generating step-by-step solutions before the final answer. Cobbe et al. [16] extend this work by constructing a much larger dataset to finetune a pre-trained large language model to solve math word problems, and a outcome-supervised verifier to rank candidate solutions. Wei et al. [79] demonstrate that the reasoning ability of a language model can be elicited through the use of prefixed rationales. Subsequent research [36, 83, 39, 96, 84] in tasks requiring human-level reasoning skills has also highlighted the efficacy of rationale augmentation.

Among all the reasoning tasks, we select mathematical reasoning to evaluate easy-to-hard generalization ability, given that mathematical reasoning serves as a valuable assessment for complex reasoning abilities and features a clear delineation of difficulty levels. Recent research efforts focus on prompt design [79, 95, 24, 90, 94] to elicit the intrinsic reasoning capabilities of models, or data engineering for fine-tuning [44, 87, 88, 86, 26, 42, 2, 6], which draws on experts to provide high-quality training datasets. Our work is categorized as fine-tuning based work. However, unlike previous work, our focus lies in exploring how to generalize to more challenging mathematical problems when only provided with easy mathematical data.

## A.2 Outcome Reward Models & Process Reward Models

For some multi-step complex reasoning tasks, such as generating highly complex code, it may be challenging for humans to fully grasp the outputs produced by an advanced AI system. In such scenarios, process-supervised reward models (PRMs) present a promising solution [74, 40]. These models operate by supervising each step in the reasoning or generation process, rather than focusing solely on the end result. They are particularly effective in tasks where the reasoning process itself is as important as the final outcome [32, 33].

Uesato et al. [74] find that process-supervised reward models (PRMs) achieve better performance than outcome-supervised reward models (ORMs) when re-ranking sampled solutions from the policy model, but their performance is similar during reinforcement learning (RL) via expert iteration [66, 4, 54, 89, 28, 67]. Lightman et al. [40] compare ORMs and PRMs with a more capable base model [48] and significantly more human-labeled process feedback on the more challenging MATH dataset, and also find that PRMs significantly outperform ORMs in the reranking setting. In contrast to these works, which only study the effectivenss of PRM in an independent and identically distributed (IID) domain, we study the utilization of PRMs in the easy-to-hard generalization scenario, and show that easy-to-hard evaluators instantiated by PRMs can enable easy-to-hard generation of policy models.

# B Reinforcement Learning Algorithms

**Reinforced Self-Training (ReST)** is an offline RL algorithm, which alternates between generating samples from the policy, which are then used to improve the LLM policy with RM-weighted SFT [28, 67]. Its variants include expert iteration [4] and rejection sampling fine-tuning [72, 87].

**Direct Policy Optimization (DPO)** is a class of offline RL algorithms [56] that consider both positive and negative gradient updates. It fine-tunes the policy model on a preference dataset consisting of paired positive and negative samples. The variants include NLHF [45], IPO [5], and SLiC [91, 92]. Recent work shows that iteratively applying DPO leads to improved performance [82].

**Proximal Policy Optimization (PPO)** is an online RL algorithm which samples from the policy during fine-tuning [62]. It is widely used in RLHF [68, 7, 50] and RLAIF [8, 69].

# C Hyper-parameters

## C.1 Supervised Fine-Tuning & Reward Modeling

For the PRM800K dataset [40], the SFT model is trained using steps that are labeled as correct. For the MetaMath dataset [86], given that the original dataset can contain upwards of ten solutions for the same question, potentially leading to over-fitting, we implement a filtering process. This process ensures that, during any given epoch, no more than three solutions per question are retained, thereby mitigating the risk of over-fitting.

The PRMs are trained on the corresponding released dataset [40, 75]. For generating solutions to train ORMs, we sample 32 solutions for each question from the language model using top-K sampling with K=20 and temperature of 0.7. We also ensure that the ratio between positive and negative samples for each question is between 1:3 to 3:1.

See Table 4 for a list of training hyper-parameters used in the training jobs. We use full fine-tuning for all SFT/RM training.

Table 4: Hyper-parameters in our SFT/RM training jobs

|  |  | PRM800K | | | | METAMATH | |
|  |  | SFT | PRM | ORM | OPRM | SFT | PRM |
| --- | --- | --- | --- | --- | --- | --- | --- |
| LLEMMA-7B | LEARNING RATE | 2E-5 | 2E-5 | 2E-5 | 2E-5 | 8E-6 | 2E-5 |
|  | EPOCHS | 3 | 2 | 2 | 2 | 3 | 2 |
|  | BATCH SIZE | 128 | 128 | 128 | 128 | 128 | 128 |
|  | MAX SEQ LEN | 768 | 768 | 1024 | 1024 | 1024 | 768 |
|  | DTYPE | BF16 | BF16 | BF16 | BF16 | FP32 | BF16 |
| LLEMMA-34B | LEARNING RATE | 1E-5 | 1E-5 | 1E-5 | 1E-5 | 5E-6 | - |
|  | EPOCHS | 3 | 2 | 2 | 2 | 3 | - |
|  | BATCH SIZE | 128 | 128 | 128 | 128 | 128 | - |
|  | MAX SEQ LEN | 768 | 768 | 1024 | 1024 | 768 | - |
|  | DTYPE | BF16 | BF16 | BF16 | BF16 | FP32 | - |

## C.2 Re-Ranking

For majority voting, weighted voting, and best-of-$n$, we sample from the language model using top-K sampling with K=20 and temperature of 0.7. At test time, we use the ORM's prediction at the final token as the overall score for the solution, and use the PRM's prediction at each intermediate step (denoted by the new line symbol) and the final token as the process reward scores.

## C.3 Reinforcement Learning

We use full fine-tuning during the RL stage.

**ReST-EM** Following Singh et al. [67], we sample 32 solutions for each question from the language model using top-K sampling with K=40. We also used a cut-off threshold of 10 for the maximum number of solutions per problem [89, 67]. We performed iterative ReST training for two epochs, and observed performance degeneration starting from the third epoch. For PRM800K, we used a temperature of 1.0, while for MetaMath, we used a temperature of 1.2. The rest training hyper-parameters are the same as in SFT training.

**Iterative DPO** We sample 8 solutions for each question from the language model using top-K sampling with K=20 and temperature of 1.0. We use the process reward model to assign a score between 0 and 1 to each solution, and use final-answer reward to assign an additional 0/1 score to each solution. A preference training pair is constructed only when the score difference between positive and negative solutions is greater than 1.0. We used a cut-off threshold of 3 for the maximum number of preference pairs per problem.

Table 5: Full results of comparing reinforcement learning (RL) approaches for easy-to-hard generalization. All methods are of 7b size and evaluated with greedy decoding. † indicates the model is trained with additional final-answer labels on hard tasks (similar to Singh et al. [67]), which is not strictly a easy-to-hard generalization setup.

| | RL DATA | REWARD | | ACCURACY | | |
| | | FINAL-ANSWER | PROCESS RM | EASY (LEVEL 1-3) | HARD (LEVEL 4-5) | ALL |
|---|---|---|---|---|---|---|
| *(SFT / PRM trained on level 1-3 of PRM800K)* | | | | | | |
| SFT | | | | 28.2 | 12.2 | 19.8 |
| ReST-EM | EASY | EASY | × | 33.2 | 12.6 | 22.4 |
| ReST-EM | HARD | HARD | × | 31.9 | 8.0 | 19.4 |
| ReST-EM† | ALL | ALL | × | 35.7 | 8.8 | 21.6 |
| ITERATIVE DPO | EASY | EASY | √ | 42.0 | 12.2 | 26.4 |
| ITERATIVE DPO† | ALL | ALL | √ | 38.2 | 11.5 | 24.2 |
| PPO | EASY | EASY | × | 42.0 | 14.1 | 27.4 |
| PPO | HARD | HARD | × | 34.0 | 9.2 | 21.0 |
| PPO† | ALL | ALL | × | 42.0 | 10.7 | 25.6 |
| PPO | ALL | EASY | √ | **45.4** | **14.9** | **29.4** |
| *(SFT / PRM trained on level 1-5 of MetaMath / Math-Shepherd)* | | | | | | |
| LLEMMA-BASED SFT SoTA (OURS) | | | | 51.7 | 13.7 | 31.4 |
| PREVIOUS RL SoTA [75] | | | | - | - | 33.0 |
| *(SFT / PRM trained on level 1-3 of MetaMath / Math-Shepherd)* | | | | | | |
| SFT | | | | 44.1 | 14.9 | 28.8 |
| ReST-EM | EASY | EASY | × | 50.4 | 14.5 | 31.6 |
| ITERATIVE DPO | EASY | EASY | √ | **53.8** | **16.0** | **34.0** |
| ITERATIVE DPO | ALL | EASY | √ | 49.6 | 10.7 | 29.2 |
| ITERATIVE DPO† | ALL | ALL | √ | 47.9 | 12.2 | 29.2 |
| PPO | EASY | EASY | × | 50.8 | 15.3 | 32.2 |
| PPO† | ALL | ALL | × | 50.8 | 13.4 | 31.2 |
| PPO | ALL | EASY | √ | **53.8** | **16.0** | **34.0** |

For all DPO training [56], we used a learning rate of $2 \times 10^{-6}$, a batch size of 64, and a DPO training epoch of 1. We set $\beta = 0.1$ for all DPO experiments, and performed at most 5 DPO iterations (i.e., sampling new solutions and performing one DPO epoch).

**PPO** We follow Dubois et al. [22] on the implementation of the PPO algorithm, which is a variant of [50][6]. Specifically, we normalize the advantage across the entire batch of rollouts obtained for each PPO step and initialize the value model from the reward model.

We clipped the gradient by its Euclidean norm at a limit of 1. Our training spanned 500 PPO steps on the RL data (MATH questions except MATH500 and our 500 validation questions). For generalized advantage estimation (GAE; Schulman et al. [61]), both $\lambda$ and $\gamma$ were set at 1.

For PRM800K, we used a batch size of 512 for each PPO step. This comprised 8 epochs of gradient steps, each having 64 rollouts. We applied a peak learning rate of $2 \times 10^{-5}$ with cosine decay. We opted for a constant KL regularizer coefficient of 0.01, and a sampling temperature of 0.7.

For MetaMath/Math-Shepherd, we used a batch size of 512 for each PPO step. This comprised 2 epochs of gradient steps, each having 256 rollouts. We applied a peak learning rate of $5 \times 10^{-6}$ with cosine decay. We opted for a constant KL regularizer coefficient of 0.002, and a sampling temperature of 1.2.

## D    Re-ranking Results on MetaMath

Similar to Sec. 4.2.1, we assess the effectiveness of process reward models on the MetaMath/Math-Shepherd dataset [86, 75]. From Figure 6, we can see that PRMs are also more effective on harder tasks when trained on MetaMath/Math-Shepherd.

---

[6]https://github.com/openai/lm-human-preferences

Table 6: Pass@N scores (upper bound of Best-of-N) on coding problems (APPS).

| | SFT / ORM TRAIN DATA | DECODING | ACCURACY (%) | | | |
| --- | --- | --- | --- | --- | --- | --- |
| | | | INTRO. | INTER. | COMP. | ALL |
| CODE LLAMA - 7B | EASY | PASS @ 1 | 11.0 | 1.6 | 0.0 | 3.2 |
| | EASY | PASS @ 4 | 20.0 | 4.7 | 2.0 | 7.2 |
| | EASY | PASS @ 16 | 39.0 | 9.3 | 6.0 | 14.6 |
| CODE LLAMA - 34B | EASY | PASS @ 1 | 19.0 | 5.3 | 2.0 | 7.3 |
| | EASY | PASS @ 4 | 32.0 | 10.7 | 3.0 | 13.4 |
| | EASY | PASS @ 16 | 44.0 | 18.3 | 8.0 | 21.4 |

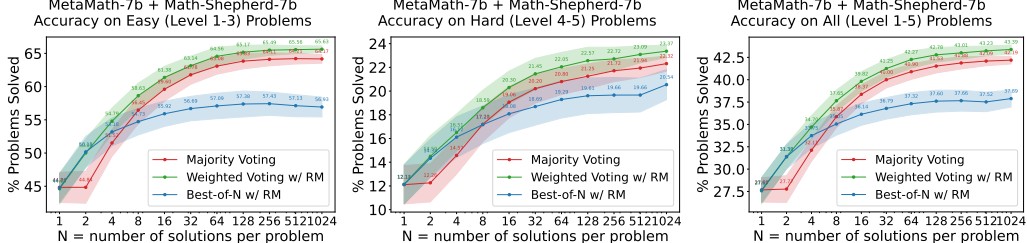

Figure 6: Easy-to-hard generalization of evaluators on MetaMath. Both SFTs and RMs are trained on the easy data.

# E Qualitative Comparison

| QUESTION | The value of $y$ varies inversely as $\sqrt{x}$ and when $x = 24$, $y = 15$. What is $x$ when $y = 3$? |
|---|---|
| METAMATH-SFT | If $y$ varies inversely as $\sqrt{x}$, then $y = \frac{k}{\sqrt{x}}$ for some constant $k$. 

 Plugging in the given values, we have $15 = \frac{k}{\sqrt{24}}$, or $k = \frac{15\sqrt{24}}{24} = \frac{15\sqrt{6}}{4}$. 

 Now we can find $x$ when $y = 3$ by plugging in these values into the equation $y = \frac{k}{\sqrt{x}}$. 
 We have $3 = \frac{\frac{15\sqrt{6}}{4}}{\sqrt{x}}$, or $3 = \frac{15\sqrt{6}}{4\sqrt{x}}$. 
 Cross-multiplying, we get $3 \cdot 4\sqrt{x} = 15\sqrt{6}$, or $12\sqrt{x} = 15\sqrt{6}$. 
 Dividing both sides by 12, we find $\sqrt{x} = \frac{15\sqrt{6}}{12\sqrt{6}} = \frac{5}{2}$. 
 Squaring both sides, we get $x = \left(\frac{5}{2}\right)^2 = \boxed{6.25}$. |
| METAMATH-PPO | If $y$ varies inversely as $\sqrt{x}$, then we can write the equation as $y = \frac{k}{\sqrt{x}}$ for some constant $k$. 
 We are given that when $x = 24$, $y = 15$, so we can substitute these values into the equation to solve for $k$: $15 = \frac{k}{\sqrt{24}}$ 
 Multiplying both sides by $\sqrt{24}$ gives $15\sqrt{24} = k$ 
 Simplifying the left side gives $15 \cdot 2\sqrt{6} = k$, which further simplifies to $30\sqrt{6} = k$ 
 Now we can use this value of $k$ in the equation to find $x$ when $y = 3$: 
 $3 = \frac{30\sqrt{6}}{\sqrt{x}}$ 
 Multiplying both sides by $\sqrt{x}$ gives $3\sqrt{x} = 30\sqrt{6}$ 
 Dividing both sides by 3 gives $\sqrt{x} = 10\sqrt{6}$ 
 Squaring both sides gives $x = (10\sqrt{6})^2 = 100(6) = \boxed{600}$ |

Table 7: Qualitative comparison between MetaMath-7b-SFT and MetaMath-7b-PPO, as well as the first error detected in MetaMath-SFT greedy decoding by Math-Shepherd. Examples are cherry-picked for better illustration.

# F Performance of DPO Iterations

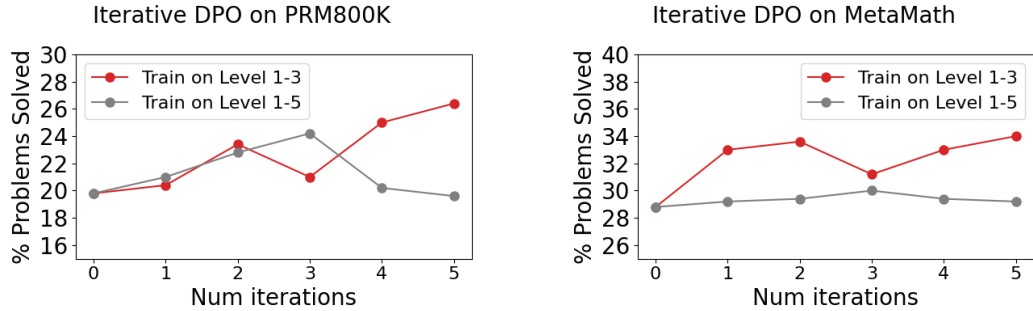

Figure 7: Test performance as a function of DPO iterations.

| QUESTION | What is the total surface area, in square inches, of a cube with a volume of 1 cubic foot? |
|---|---|
| PRM800K-SFT | The volume of a cube is given by $V = s^3$, where $s$ is the length of one side. We are given that $V = 1$ cubic foot, so $s^3 = 1$. Taking the cube root of both sides, we find that $s = 1$. The surface area of a cube is given by $A = 6s^2$, so the total surface area of this cube is $A = 6(1^2) = \boxed{6}$ square inches. # Answer 6 |
| PRM800K-PPO | So the volume of a cube is $s^3$. And we know the volume is 1. So $s^3 = 1$. So $s = 1$. The surface area of a cube is $6s^2$. So the surface area is $6 \cdot 1^2 = 6$. So the answer is $\boxed{6}$. # Answer 6 |

Table 8: The distribution shift before and after PPO training of a Llemma-7B model trained on PRM800K. We believe such a big shift could explain why does PPO work exceptionally well on PRM800K, where the policy is initialized from a weak SFT model.

# G   Re-ranking Performance Analysis of PRMs, ORMs & OPRMs

We compare the re-ranking performance of Process Reward Models (PRMs), Outcome Reward Models (ORMs), and our proposed Outcome & Process Reward Models (OPRMs). Figure 8 shows the results on 7b models and Figure 9 is on 34b models. We find that in our setting of Llemma [6] + MATH [32], PRMs and ORMs perform similarly, with PRMs slightly outperforming ORMs on hard tasks. But the OPRMs that trained on the mixed data of PRMs and ORMs significantly outperforms both of them.

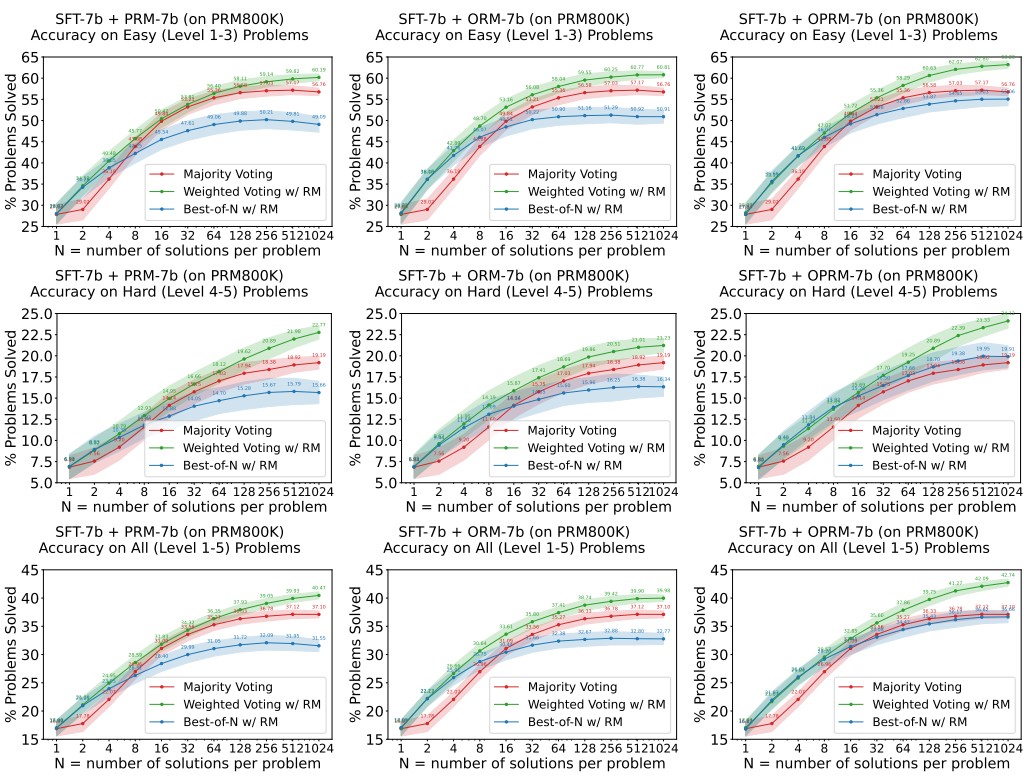

Figure 8: Comparing process reward models (PRMs, left), outcome reward models (ORMs, middle), and outcome & process reward models (OPRMs, right) on 7b models trained on the PRM800K dataset. Both SFTs and RMs are trained on the easy data.

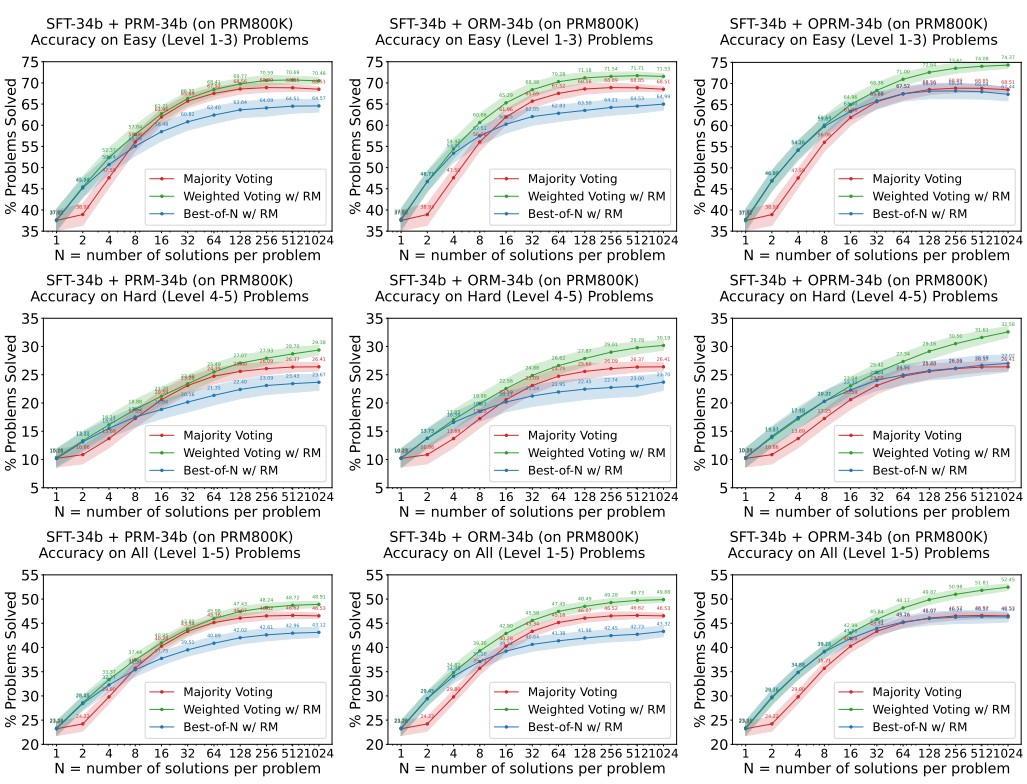

Figure 9: Comparing process reward models (PRMs, left), outcome reward models (ORMs, middle), and outcome & process reward models (OPRMs, right) on 34b models trained on the PRM800K dataset. Both SFTs and RMs are trained on the easy data.

# H Re-ranking Results on MetaMath

Similar to Sec. 4.2.1, we assess the effectiveness of process reward models on the MetaMath/Math-Shepherd dataset [86, 75]. From Figure 10, we can see that PRMs are also more effective on harder tasks when trained on MetaMath/Math-Shepherd.

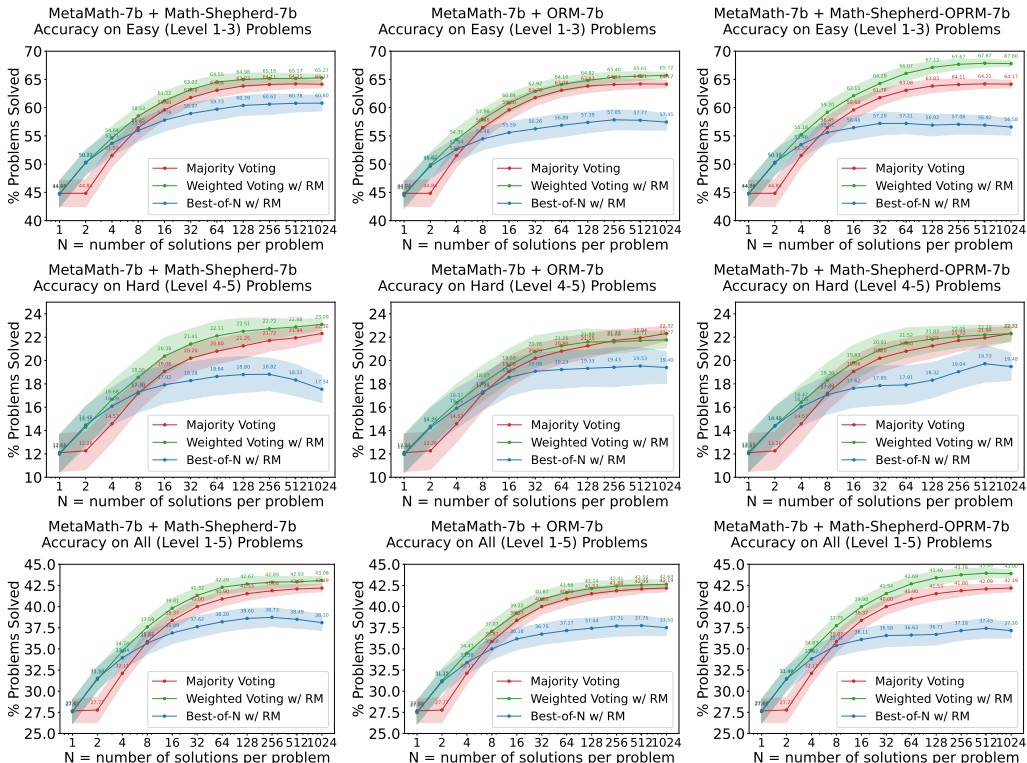

Figure 10: Comparing process reward models (PRMs, left, trained on Meth-Shepherd), outcome reward models (ORMs, middle), and outcome & process reward models (OPRMs, right) on 7b models trained on the MetaMath dataset. Both SFTs and RMs are trained on the easy data.

# I More Comparisons

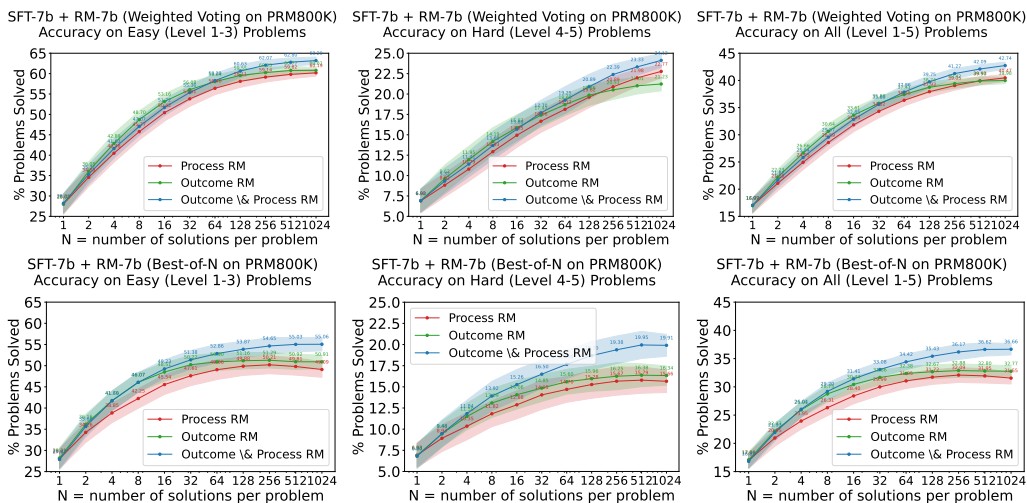

Figure 11: Comparing different reward models with Weighted Voting (upper) and Best-of-N (lower) on 7b models trained on the PRM800K dataset. Both SFTs and RMs are trained on the easy data.

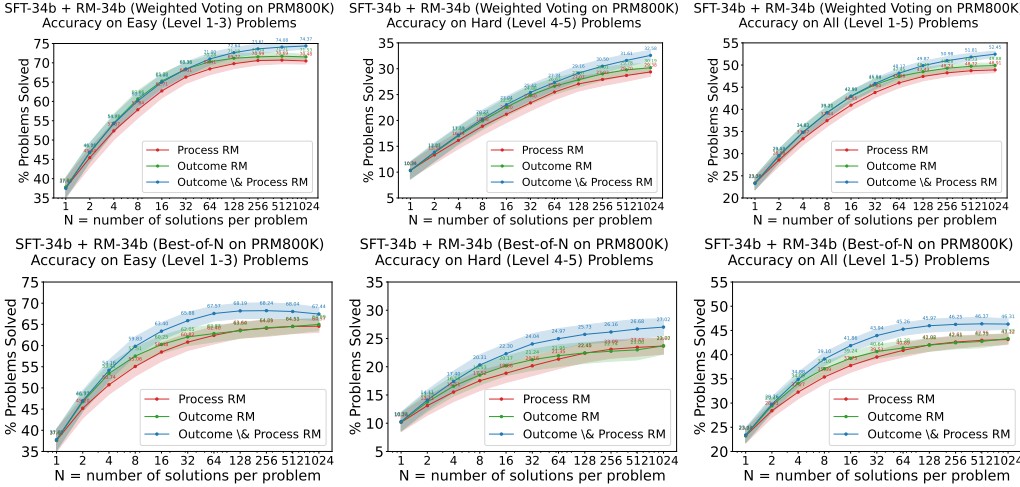

Figure 12: Comparing different reward models with Weighted Voting (upper) and Best-of-N (lower) on 34b models trained on the PRM800K dataset. Both SFTs and RMs are trained on the easy data.

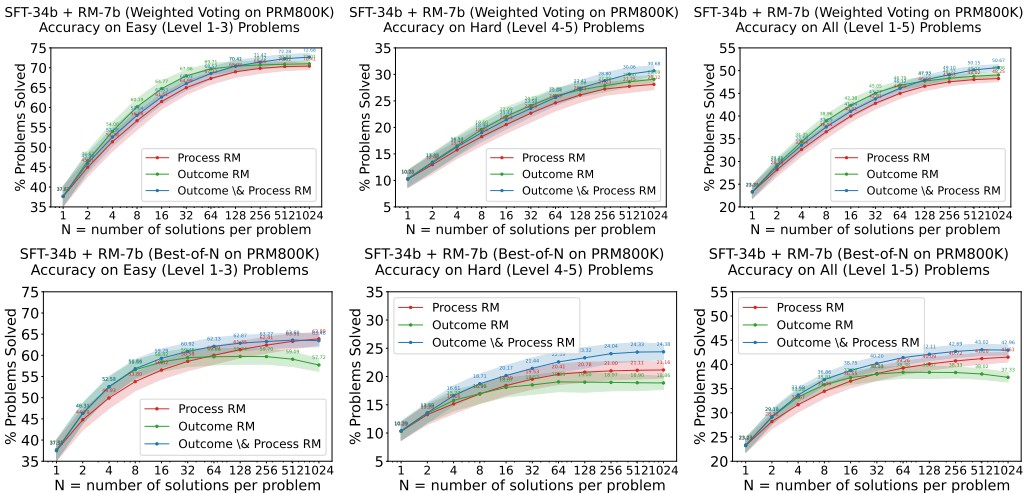

Figure 13: Comparing different reward models with Weighted Voting (upper) and Best-of-N (lower) on 34b SFT model and 7b reward model trained on the PRM800K dataset. Both SFTs and RMs are trained on the easy data.

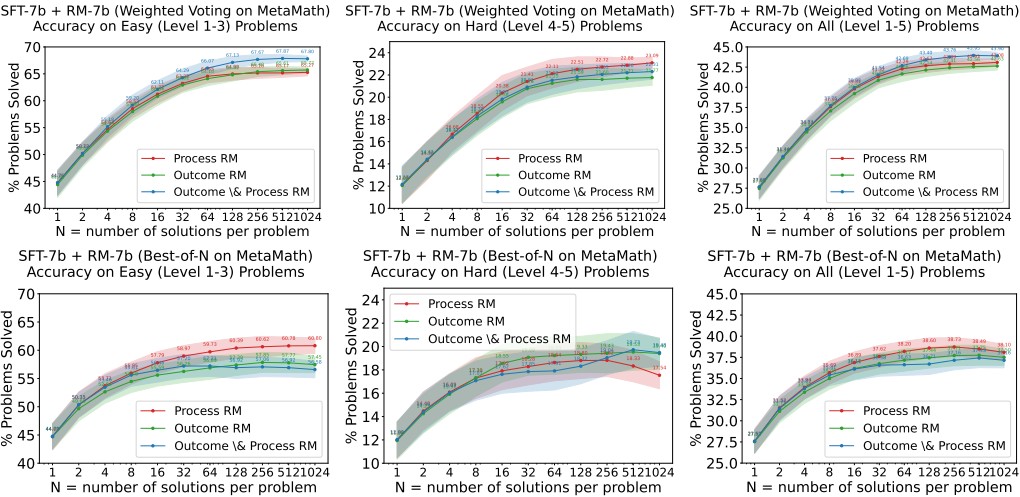

Figure 14: Comparing different reward models with Weighted Voting (upper) and Best-of-N (lower) on 7b models trained on the MetaMath dataset. Both SFTs and RMs are trained on the easy data.

Table 9: Results of Full, Easy-to-Hard, & Hard-to-Easy SFT training of the Llemma-7b model

| | TRAINING DATA | PRM800K | | METAMATH | |
| --- | --- | --- | --- | --- | --- |
| | | ALL | HARD | ALL | HARD |
| FULL SFT | ALL | **20.6** | 9.9 | **31.4** | 13.7 |
| EASY-TO-HARD SFT | EASY | 19.8 | 12.2 | 30.0 | 14.9 |
| HARD-TO-EASY SFT | HARD | 18.4 | **13.0** | 30.4 | **15.3** |

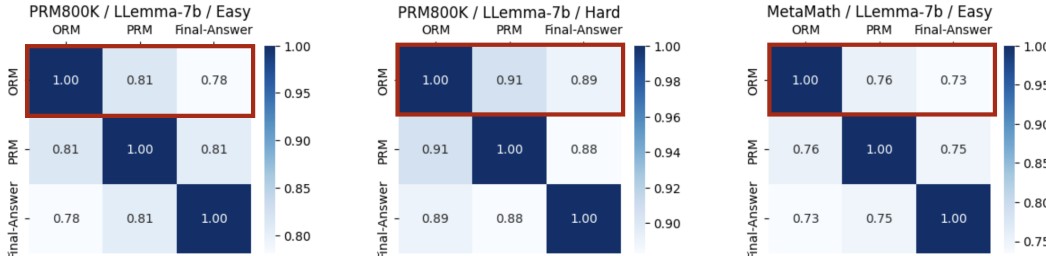

Figure 15: The agreement between the prediction from the Llemma-7b-based reward model when trained on ORM and PRM data, and their agreement to ground-truth final-answer labels.

## J Hard-to-Easy Generalization

From Table 5, it is evident that reinforcement learning training on hard tasks alone significantly underperforms compared to training the model on easy tasks or on all tasks. This difference is especially pronounced for PPO on the PRM800K dataset. This raises a crucial question: does training on hard tasks only generalize to easy tasks?

To address this, we fine-tuned the Llemma-7b model using all data (easy and hard), only easy data, and only hard data. As shown in Table 9, we found that training on all data consistently yields the best performance. Conversely, the generator's performance deteriorates when transitioning from easy-to-hard and hard-to-easy tasks. This suggests that language models face difficulties in generalizing in both directions.

It is also worth noting that while Full SFT underperforms Easy-to-Hard SFT and Hard-to-Easy SFT on hard test questions, it eventually outperforms Easy-to-Hard SFT and Hard-to-Easy SFT when evaluated on all test questions. We believe that this is because by exposing the model to a wider variety of unique questions and difficulties, it gains a better understanding of the problem space in general, as measured by the accuracy on the full distribution.

## K On ORM's Approximation of PRM Labels

From Sec. G, we observe that in PRM800K, PRMs and ORMs exhibit similar performance levels, with OPRMs outperforming both. This raises the question of why ORMs also demonstrate strong easy-to-hard generalization ability. A straightforward explanation is that ORMs are trained to approximate PRM labels [74]. Specifically, ORMs are trained to predict the correctness of the entire solution through value estimation. As Uesato et al. [74] state, "it is simpler for the ORM to learn to recognize when steps are correct than it is to check the answer by internally computing the final answer itself."

Nevertheless, people may argue that the conclusion from Uesato et al. [74] is based on GSM8K's experimental results, so the conclusion may not transfer to the more challenging Hendrick's MATH dataset. To show the universal existence of "ORM's approximation of PRM labels", we further conduct evaluation of agreement between different rewards on two variants of the MATH dataset: PRM800K and MetaMath.

The results are shown in Figure 15. Similarly to the findings from Uesato et al. [74], we see that the ORM has higher agreement with the PRM, despite being trained to predict the Final-Answer rewards.

Thus, "this result indicates that the ORM tends more towards predicting whether the full trace is correct, and not just whether the final answer is correct."

Overall, this shows easy-to-hard generalization is not exclusively linked to reward models trained on explicit step-wise annotations. It also applies to ORMs that are trained to perform value estimation and practically evaluates each solution step.

We also perform DPO training on a MetaMath-initialized Llemma-7b model. We find that in this RL setting, re-ranking the output pairs with ORM also gives similar performance to re-ranking with PRM (29.2 v.s. 30.4 & 31.2 v.s. 34.0).

## L    Analysis of Aggregation Functions in PRMs & OPRMs

We explored different methods to consolidate step-wise prediction scores into a single score value, a process we describe as employing an aggregation function, during the use of the evaluator. Lightman et al. [40] report comparable performance when using $\texttt{min}$ (minimum) and $\texttt{prod}$ (product) as the aggregation function to reduce multiple scores into a single solution-level score. Note that when training PRMs on PRM800K [40], we have already considered neutral steps to be positive as training labels.

Following Wang et al. [77], given $\{p_1, p_2, \ldots, p_n\}$ as a list of predicted correctness probability of each step (including the final answer), we considered the following aggregation functions:

$$\texttt{min} = \min\{p_1, p_2, \ldots, p_n\} \tag{1}$$

$$\texttt{max} = \max\{p_1, p_2, \ldots, p_n\} \tag{2}$$

$$\texttt{prod} = \prod_i p_i \tag{3}$$

$$\texttt{mean} = \frac{\sum_i p_i}{n} \tag{4}$$

$$\texttt{mean\_logit} = \sigma\left(\frac{\sum_i \log \frac{p_i}{1-p_i}}{n}\right) \tag{5}$$

$$\texttt{mean\_odd} = \text{ReLU}\left(\frac{\sum_i \frac{p_i}{1-p_i}}{n}\right) \tag{6}$$

$$\texttt{last} = p_n \tag{7}$$

In Figure 16-18, we perform analysis of aggregation functions on PRM800K and Math-Shepherd (from MetaMath) datasets with weighted voting and best-of-$n$ decoding and PRMs or OPRMs. In general, we find $\texttt{prod}$ works universally well in weighted voting and $\texttt{min}$ works well in best-of-$n$. So we adopt these two strategies in our main experiments.

One interesting finding is that for reward models trained on the human annotated process reward (e.g., PRM800K), the $\texttt{last}$ strategy does not perform very well, but $\texttt{last}$ works much better on OPRMs and pseudo PRMs (e.g., Math-Shepherd). This could partially explain why OPRMs does not further improve the performance on the Math-Shepherd dataset.

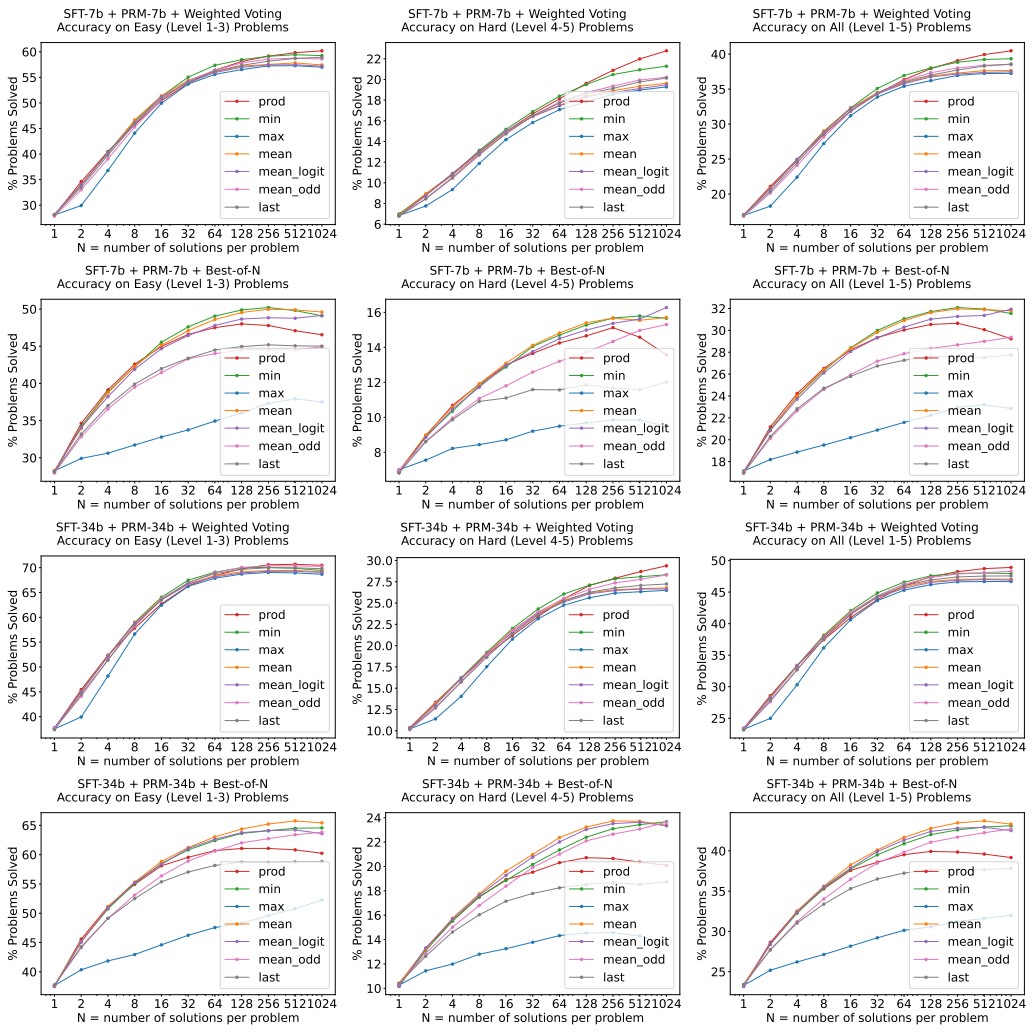

Figure 16: Analysis of aggregation functions in process reward models (PRMs) on the PRM800K dataset with Weighted Voting and Best-of-N. Both SFTs and RMs are trained on the easy data.

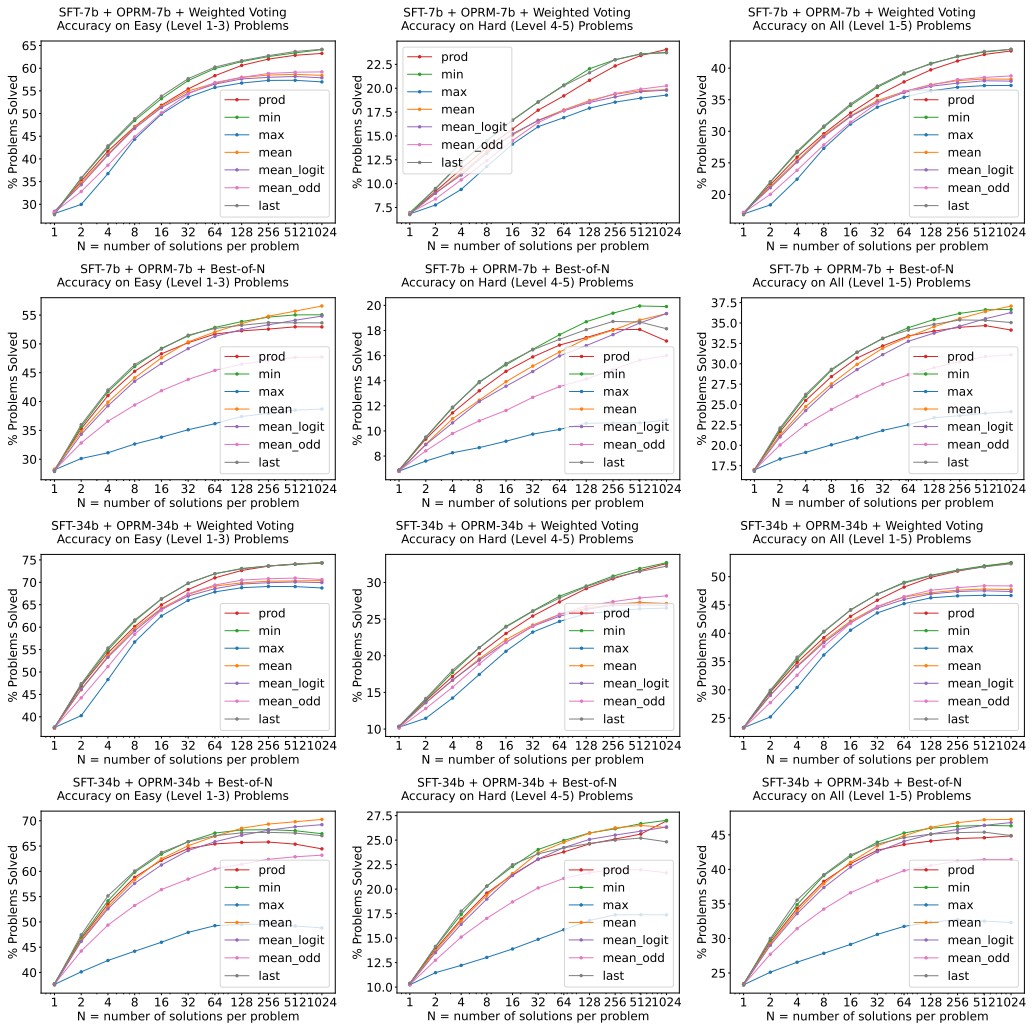

Figure 17: Analysis of aggregation functions in outcome & process reward models (OPRMs) on the PRM800K dataset with Weighted Voting and Best-of-N. Both SFTs and RMs are trained on the easy data.

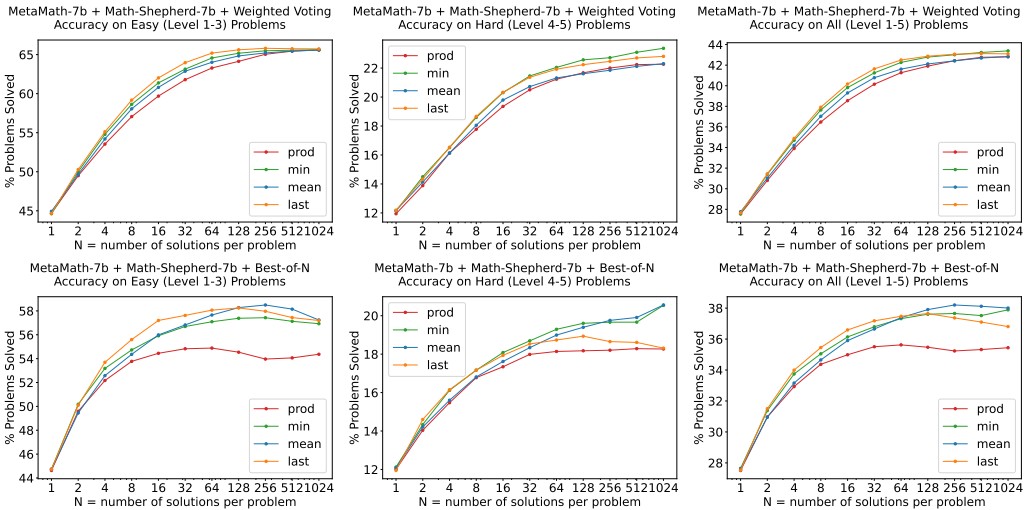

Figure 18: Analysis of aggregation functions in psuedo process reward models (PRMs) on the Math-Shepherd (from MetaMath) dataset with Weighted Voting and Best-of-N. Both SFTs and RMs are trained on the easy data.

# M  Societal Impact

Our work on easy-to-hard generalization has the potential for both positive and negative societal impacts. On the positive side, this approach could enable AI systems to tackle increasingly complex problems in domains such as scientific discovery, healthcare, and education, potentially leading to groundbreaking advancements that benefit society. However, the development of AI systems that can operate beyond human supervision also raises concerns about the transparency, accountability, and potential misuse of such systems. It is crucial to carefully consider the ethical implications and establish robust safeguards to mitigate the risks of unintended consequences or malicious applications. Ongoing research and public discourse on the responsible development and deployment of these technologies will be essential to ensure that their societal benefits outweigh the potential drawbacks.

# N    Fine-Grained Analysis of OPRMs' Re-ranking strategies

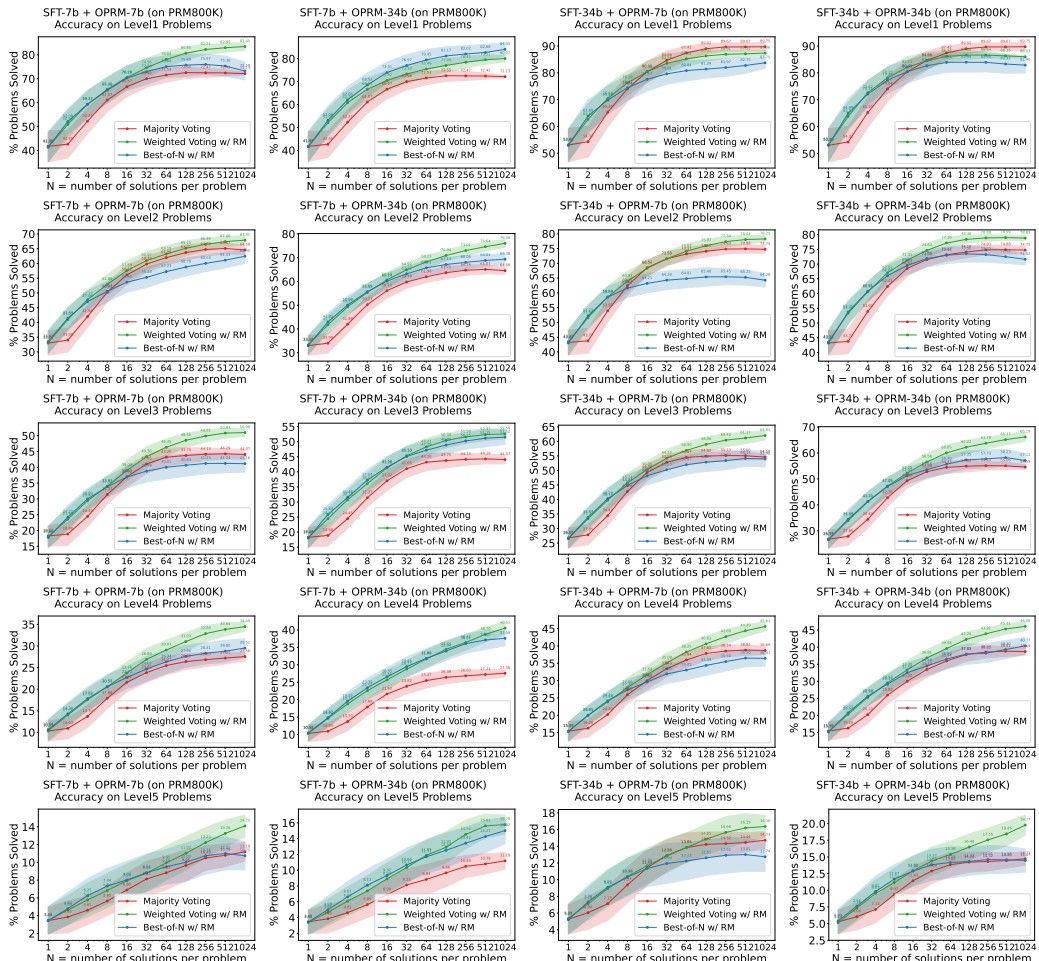

Figure 19: Easy-to-hard generalization for different difficulty levels' data. Both SFTs and OPRMs are trained on the level1-3 data. Each row compares the performance of different OPRMs' reranking strategies across different levels' data.

As shown in Figure 19, both the Best-of-N and Weighted Voting strategies demonstrate strong performance across all levels, which leverage the advantages of OPRM methods and thereby underscoring OPRM's effectiveness. Furthermore, despite the SFT models and OPRM models being trained on level 1-3 data, re-ranking strategies enhanced by OPRMs continue to perform well on the unseen and more challenging level 4-5 data. This indicates the feasibility of generalizing from easier to harder tasks using OPRMs.

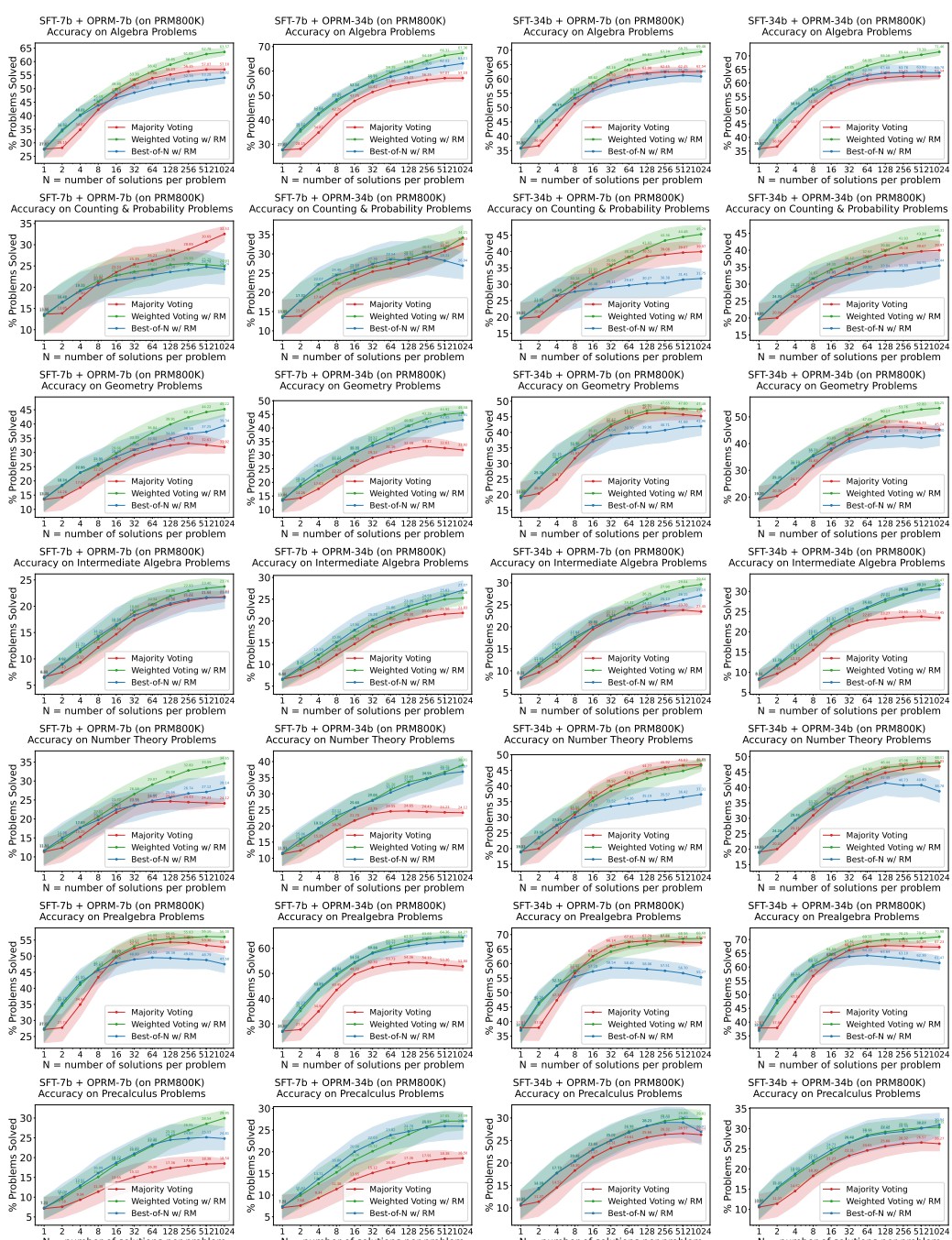

Figure 20: Easy-to-hard generalization for different type's data. Both SFTs and OPRMs are trained on the easy data. Each row compares the performance of different OPRMs' reranking strategies across different types' data.

To determine which types of data benefit more from OPRM's easy-to-hard generalization and which types still struggle with this challenging generalization, we compare OPRM's generalization abilities on different problem types in Figure 20. Among Algebra, Counting & Probability, Geometry, Intermediate Algebra, Number Theory, Prealgebra, and Precalculus problems, OPRMs generalize best on Algebra, Intermediate Algebra, and Precalculus problems. Conversely, OPRMs generalize worst on Counting & Probability problems. These findings are highly valuable for practical system design, allowing us to decide when to use OPRMs to enhance performance based on the downstream data type.

# O  Pass@N Analysis for Different Math Questions

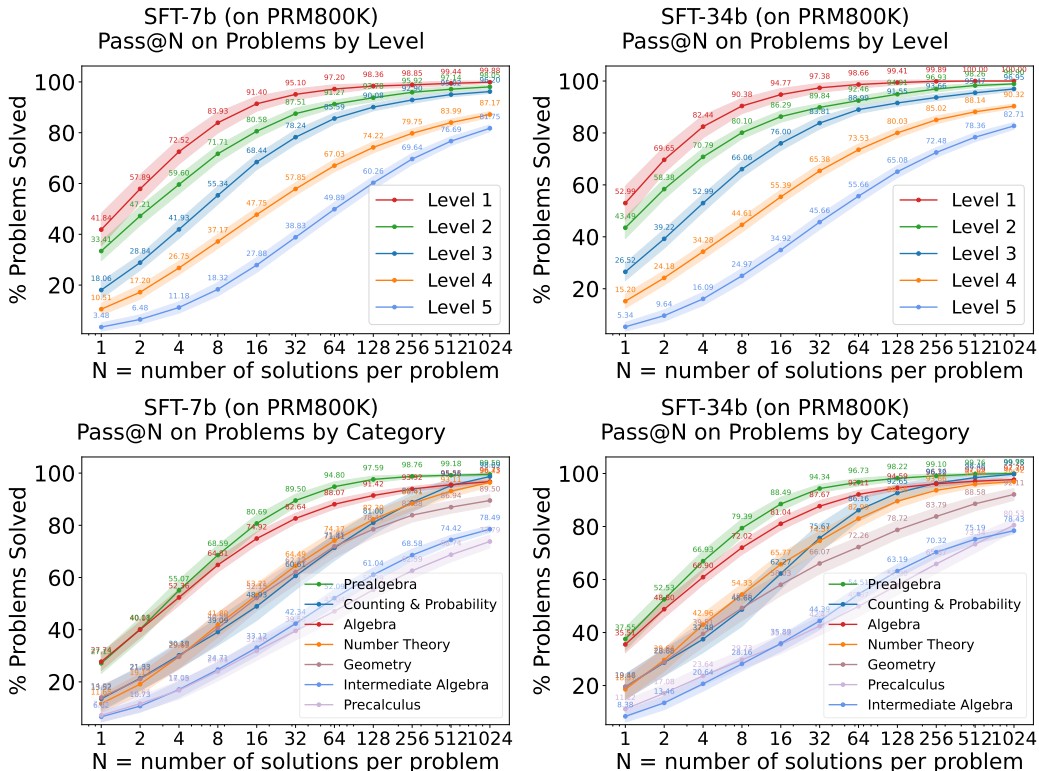

Figure 21: The increase of Pass@N metrics with the number of samples, divided by level and category. As the number of samples increases, the Pass@N rate tends to improve. However, not all problems will eventually yield a correct solution, meaning that even as N approaches infinity, the Pass@N metric may not reach 100%.

Figure 21 shows the changes in Pass@N as the number of samples increases. As shown in the figure, not all problems reach a 100% Pass@N rate, even with a larger N. Specifically, three key observations can be made: First, Pass@N is strongly correlated with problem difficulty. As depicted in Figure 21, for Level 1 problems, the Pass@N rate nearly saturates with more samples, while for Level 5 problems, many instances still do not achieve a correct solution. Second, larger models (e.g., 34b) perform significantly better on Pass@1 compared to smaller models (e.g., 7b). However, as N increases, the advantage of the larger model diminishes on metrics like Pass@1024. Third, the variations in Pass@N across different categories of math questions are different, which is expected, as the difficulty of different types of math questions varies for large language models.

# P Case Studies for OPRM

**Case Study 1**: A robot moving forward at a constant speed takes 2.5 hours to travel 1 kilometer. Moving forward at this same constant speed, it takes the robot 90 seconds to travel the length of a particular hallway. How many meters long is the hallway? **Level: 5 GT: 10**

**Voting V.S. Reward**: {Answer: 36, Voting times: 139, OPRM reward: 11.3}, {Answer: 10, Voting times: 99, OPRM reward: 23.7}

| **Majority Solution:** | #Reward | **OPRM Solution:** | #Reward |
|---|---|---|---|
| Let's call the number of meters in the hallway x. | 0.97 | To solve this problem, I need to find the relationship between the robot's speed, distance, and time. | 0.98 |
| We know the robot moved 1 km in 2.5 hours. | 0.97 | I can use the formula distance = speed * time to write an equation for each case. | 0.98 |
| That means it moved x meters in 90 seconds. | 0.28 | Let s be the robot's speed in meters per hour. | 0.98 |
| We can write that as the equation $\frac{1}{2.5}=\frac{x}{90}$. | 0.11 | Then the first case gives me 1000 = s * 2.5, or s = 1000 / 2.5 = 400 meters per hour. | 0.98 |
| To solve for x, we can multiply both sides by 90 to get $36=x$. | 0.07 | The second case gives me the length of the hallway = s * 90 / 3600, or length = 400 * 90 / 3600 = 10 meters. | 0.79 |
| So the hallway is 36 meters long. | 0.03 | | |
| # Answer | | # Answer | |
| 36       **#Final Reward: 6.7e-5** | | 10       **#Final Reward: 0.74** | |

Figure 22: Case Study 1. The evaluator can help identify and reduce the confidence of hallucinations caused by misleading information.

**Case Study 2**: In a regular polygon, the measure of an interior angle is 6.5 times the measure of an exterior angle. How many sides does the polygon have? **Level: 4 GT: 15**

**Voting V.S. Reward**: {Answer: 0, Voting times: 132, OPRM reward: 0.03}, {Answer: 15, Voting times: 94, OPRM reward: 17.3}

| **Majority Solution:** | #Reward | **OPRM Solution:** | #Reward |
|---|---|---|---|
| So we know that the measure of an interior angle is 6.5 times the measure of an exterior angle. | 0.98 | I know that the sum of the interior angles of any polygon with n sides is (n-2) times 180 degrees, and the sum of the exterior angles is 360 degrees. | 0.98 |
| And we know that the sum of the measures of the interior angles of a polygon is given by the formula (n-2)180, where n is the number of sides. | 0.89 | I can write an equation relating the interior and exterior angles of a regular polygon: (n-2) * 180 = 6.5 * 360, where n is the number of sides. | 0.84 |
| Right. So we can say that 6.5(n-2)=(n-2)180. | 0.22 | I can simplify this equation by dividing everything by 180: n - 2 = 6.5 * 2, which gives me n - 2 = 13. | 0.97 |
| And that simplifies to 6.5=180, which is false. | 0.12 | I can add 2 to both sides to isolate n: n = 13 + 2, which gives me n = 15. | 0.95 |
| So there is no such polygon. | 0.03 | # Answer | |
| # Answer | | 15 | |
| 0       **#Final Reward: 6e-4** | |       **#Final Reward: 0.75** | |

Figure 23: Case Study 2. The evaluator can assist in reducing the confidence of solutions that misuse mathematical theorems or formulas.

We have included more case studies in Figures 22 and 23. We find that evaluator can help generalize to harder ones in the following ways:

- The evaluator can help identify and reduce the confidence of hallucinations caused by misleading information in problems. As demonstrated in Case Study 1, the solution selected by majority voting with an answer of 36 is misled by the different units of measurement in the problem (2.5 hours and 90 seconds), resulting in an incorrect solution. Then, the ORPM model successfully gives this solution a low score.

- The evaluator can assist in reducing the confidence of solutions that misuse mathematical theorems. In Case Study 2, the majority solution incorrectly applies the theorem "the sum of the exterior angles of a polygon is 360°", leading to erroneous reasoning, and low confidence by the ORPM model.

# Q   Step and Outcome ROC Curve

To assess the accuracy of reasoning step judgments of different reward models, we conducted additional experiments using the PRM800K-test data, which includes correctness annotations for each step, to test our model's ability to distinguish correct reasoning steps. We randomly selected a portion of PRM800K-test data to balance positive and negative samples. The accuracy of the reasoning steps for the three models is shown in Table 10. This

| Model | Step ACC (%) | Outcome ACC (%) |
|---|---|---|
| ORM-PRM800K-7B | 64.3 | 71.7 |
| ORM-PRM800K-7B | **80.4** | 63.5 |
| OPRM-PRM800K-7B | 79.8 | **74.4** |

Table 10: The accuracy of the reasoning steps for different models.

table demonstrates the effectiveness of our trained PRM, showing that PRM has a significantly greater ability to distinguish steps compared to ORM. Additionally, in Figure 24, we present the Step ROC curves of three models, where PRM and OPRM exhibit better step discrimination abilities compared to ORM. However, it is important to note that a stronger ability to distinguish steps does not necessarily indicate that the evaluator is more helpful for generation. We then also present the Outcome ROC curves of three models on discriminating the final outcome. We collect data generated on MATH500 test set from our 7B policy model. According to the final outcome and groundtruth, we label each data and select a positive-negative balanced set to plot the Outcome ROC curves, where OPRM exhibits better outcome discrimination abilities compared to ORM and PRM. The above table also shows the effectiveness of OPRM on Outcome discrimination ability.

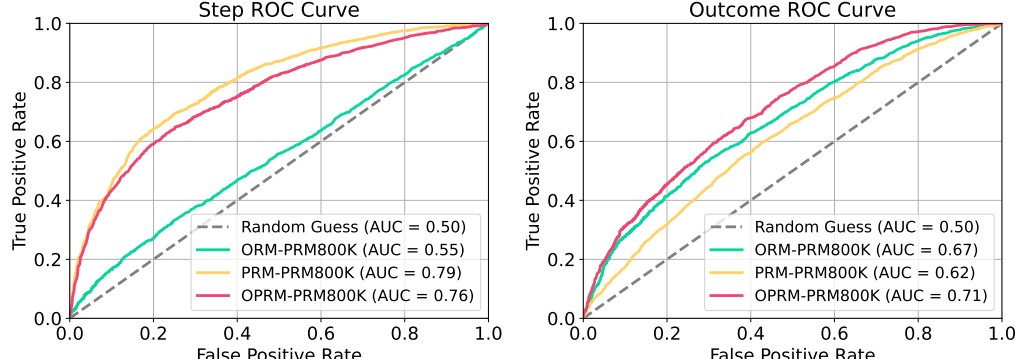

Figure 24: The Step and Outcome ROC Curve show the ability of discriminate the reasoning step and the whole reasoning solution, illustrating OPRM's effectiveness on identifying both steps and outcomes.

# R  Few-Shot Prompt in In-Context Learning

We sample with temperature T = 0.9 for ICL-related experiments. We list our few-shot sample for the In-context learning experiments in Table 1.

**4-shot example for PRM800K Full ICL:**

```
Problem:
In right triangle $PQR$, we have $\\angle Q = \\angle R$ and $PR = 6\\sqrt{2}$.
    What is the area of $\\triangle PQR$?

Solution:
I see that this is an isosceles right triangle, since $\\angle Q = \\angle R =
    45^\\circ$.\n\nThat means that $PQ = PR = 6\\sqrt{2}$.\n\nTo find the area of a
     triangle, I need to multiply the base and the height and divide by 2.\n\nIn
    this case, the base and the height are both $6\\sqrt{2}$, so the area is $(6\\
    sqrt{2})(6\\sqrt{2})/2$.\n\nI can simplify this expression by using the
    property that $\\sqrt{a}\\sqrt{b} = \\sqrt{ab}$.\n\nSo, the area is $(\\sqrt
    {2}\\sqrt{2})(6\\cdot 6)/2 = 2\\cdot 36/2 = 36$.
Final Answer: The final answer is $36$. I hope it is correct.

Problem:
The length of the longer side of rectangle $R$ is $10$ percent more than the length
     of a side of square $S.$ The length of the shorter side of rectangle $R$ is
    $10$ percent less than the length of a side of square $S.$ What is the ratio of
     the area of rectangle $R$ to the area of square $S?$ Express your answer as a
    common fraction.

Solution:
Let the side of the square be $s$.\n\nThen the longer side of the rectangle is $1.1
    s$.\n\nAnd the shorter side of the rectangle is $0.9s$.\n\nThe area of the
    square is $s^2$.\n\nThe area of the rectangle is $1.1s * 0.9s = 0.99s^2$.\n\nSo
     the ratio of the area of rectangle $R$ to the area of square $S$ is $\\frac
    {0.99s^2}{s^2} = 0.99$.\n\nTherefore, the answer is $\\boxed{\\frac{99}{100}}$.
Final Answer: The final answer is $\\frac{99}{100}$. I hope it is correct.

Problem:
Compute $\\sin(-60^\\circ)$.

Solution:
I know the sine function is periodic with period $360^\\circ$, so $\\sin(-60^\\circ
    ) = \\sin(-60^\\circ + 360^\\circ) = \\sin(300^\\circ)$.\n\nI also know that
    the sine function is symmetrical about the origin, so $\\sin(-x) = -\\sin(x)$
    for any angle $x$.\n\nTherefore, $\\sin(-60^\\circ) = -\\sin(60^\\circ)$.\n\nTo
     find $\\sin(60^\\circ)$, I can use the special right triangle with angles $30
    ^\\circ$, $60^\\circ$, and $90^\\circ$ and sides in the ratio $1:\\sqrt{3}:2$.\
    n\nThe sine of an angle is the ratio of the opposite side to the hypotenuse, so
     $\\sin(60^\\circ) = \\frac{\\sqrt{3}}{2}$.\n\nTherefore, $\\sin(-60^\\circ) =
    -\\frac{\\sqrt{3}}{2}$.
Final Answer: The final answer is $-\\frac{\\sqrt{3}}{2}$. I hope it is correct.

Problem:
Simplify $\\sin (x - y) \\cos y + \\cos (x - y) \\sin y.$

Solution:
'Ok, so we have $\\sin (x - y) \\cos y + \\cos (x - y) \\sin y$\n\nNotice that $\\
    sin (x - y) \\cos y + \\cos (x - y) \\sin y$ is in the form of $\\sin(\\alpha)
    \\cos(\\beta)+\\cos(\\alpha)\\sin(\\beta)$.\n\nYes, and we know that $\\sin(\\
    alpha)\\cos(\\beta)+\\cos(\\alpha)\\sin(\\beta)=\\sin(\\alpha+\\beta)$\n\nSo, $
    \\sin (x - y) \\cos y + \\cos (x - y) \\sin y=\\sin((x-y)+y)$\n\n$\\sin((x-y)+y
    )=\\sin(x)$
Final Answer: The final answer is $\\sinx$. I hope it is correct.
```

**4-shot example for PRM800K Hard ICL:**

```
Problem:
In right triangle $PQR$, we have $\\angle Q = \\angle R$ and $PR = 6\\sqrt{2}$.
    What is the area of $\\triangle PQR$?

Solution:
I see that this is an isosceles right triangle, since $\\angle Q = \\angle R =
    45^\\circ$.\n\nThat means that $PQ = PR = 6\\sqrt{2}$.\n\nTo find the area of a
     triangle, I need to multiply the base and the height and divide by 2.\n\nIn
    this case, the base and the height are both $6\\sqrt{2}$, so the area is $(6\\
    sqrt{2})(6\\sqrt{2})/2$.\n\nI can simplify this expression by using the
    property that $\\sqrt{a}\\sqrt{b} = \\sqrt{ab}$.\n\nSo, the area is $(\\sqrt
    {2}\\sqrt{2})(6\\cdot 6)/2 = 2\\cdot 36/2 = 36$.
Final Answer: The final answer is $36$. I hope it is correct.

Problem:
The length of the longer side of rectangle $R$ is $10$ percent more than the length
     of a side of square $S.$ The length of the shorter side of rectangle $R$ is
    $10$ percent less than the length of a side of square $S.$ What is the ratio of
     the area of rectangle $R$ to the area of square $S?$ Express your answer as a
    common fraction.

Solution:
Let the side of the square be $s$.\n\nThen the longer side of the rectangle is $1.1
    s$.\n\nAnd the shorter side of the rectangle is $0.9s$.\n\nThe area of the
    square is $s^2$.\n\nThe area of the rectangle is $1.1s * 0.9s = 0.99s^2$.\n\nSo
     the ratio of the area of rectangle $R$ to the area of square $S$ is $\\frac
    {0.99s^2}{s^2} = 0.99$.\n\nTherefore, the answer is $\\boxed{\\frac{99}{100}}$.
Final Answer: The final answer is $\\frac{99}{100}$. I hope it is correct.

Problem:
Suppose that $y^3$ varies inversely with $\\sqrt[3]{z}$. If $y=2$ when $z=1$, find
    the value of $z$ when $y=4$. Express your answer in simplest fractional form.

Solution:
I know that inverse variation means that the product of the two quantities is
    constant, so I can write an equation of the form $y^3\\cdot\\sqrt[3]{z}=k$,
    where $k$ is some constant.\n\nTo find $k$, I can plug in the given values of
    $y$ and $z$: $2^3\\cdot\\sqrt[3]{1}=k$, which simplifies to $8=k$.\n\nNow I can
     use this equation to find $z$ when $y=4$: $4^3\\cdot\\sqrt[3]{z}=8$, which
    implies that $\\sqrt[3]{z}=\\frac{8}{64}=\\frac{1}{8}$.\n\nTo get rid of the
    cube root, I can cube both sides: $z=\\left(\\frac{1}{8}\\right)^3=\\frac
    {1}{512}$.
Final Answer: The final answer is $\\frac{1}{512}$. I hope it is correct.

Problem:
Let $d$ be a positive number such that when $109$ is divided by $d$, the remainder
    is $4.$ Compute the sum of all possible two-digit values of $d$.

Solution:
This problem involves finding the divisors of a given number, as well as using the
    concept of remainders.\n\nOne way to approach this is to write the division as
    a quotient and a remainder, like this: $109 = qd + 4$, where $q$ is the
    quotient.\n\nThen, I can rearrange this equation to get $105 = qd$, which means
     that $d$ is a divisor of $105$.\n\nNow, I need to find all the two-digit
    divisors of $105$.\n\nI can use prime factorization to help me do this.\n\nI
    notice that $105 = 3 \\times 5 \\times 7$, which are all prime numbers.\n\
    nTherefore, any divisor of $105$ must be a product of some combination of these
     three factors.\n\nThe possible products are $1, 3, 5, 7, 15, 21, 35, 105$.\n\
    nHowever, not all of these are two-digit numbers. Only $15, 21, 35$ are.\n\
    nThese are the only possible values of $d$ that satisfy the given condition.\n\
    nTo find their sum, I just add them up: $15 + 21 + 35 = 71$.
Final Answer: The final answer is $71$. I hope it is correct.
```

**4-shot example for PRM800K Easy-to-Hard ICL:**

---

Problem:
How many 4-letter words with at least one consonant can be constructed from the
    letters $A$, $B$, $C$, $D$, and $E$? (Note that $B$, $C$, and $D$ are
    consonants, any word is valid, not just English language words, and letters may
     be used more than once.)

Solution:
To count the number of 4-letter words with at least one consonant, I can use the
    complement principle and subtract the number of 4-letter words with no
    consonants from the total number of 4-letter words.\n\nThe total number of 4-
    letter words is simply $5^4$, since each letter has 5 choices and the order
    matters.\n\nThe number of 4-letter words with no consonants is just the number
    of ways to choose 4 letters from the 2 vowels $A$ and $E$, which is $2^4$.\n\
    nTherefore, the number of 4-letter words with at least one consonant is $5^4 -
    2^4$. 625 - 16 = 609
Final Answer: The final answer is $609$. I hope it is correct.

Problem:
Compute the integer $k > 2$ for which\n\\[\\log_{10} (k - 2)! + \\log_{10} (k - 1)!
     + 2 = 2 \\log_{10} k!.\\]

Solution:
I recognize that this equation involves logarithms of factorials, which are
    products of consecutive integers.\n\nI also know that logarithms have some
    useful properties, such as $\\log_{10} a + \\log_{10} b = \\log_{10} (ab)$ and
    $c \\log_{10} d = \\log_{10} d^c$.\n\nUsing these properties, I can simplify
    the equation as follows: \\[\\log_{10} (k - 2)! + \\log_{10} (k - 1)! + 2 = 2
    \\log_{10} k! \\implies \\log_{10} \\left[ (k - 2)! (k - 1)! 100 \\right] = \\
    log_{10} (k!)^2.\\]\n\nSince the bases of the logarithms are equal, I can
    conclude that the arguments must also be equal, i.e., \\[(k - 2)! (k - 1)! 100
    = (k!)^2.\\]\n\nNow I have a simpler equation to solve for $k$.\n\nI notice
    that the left-hand side has a factor of $(k - 2)!$, which is also a factor of $
    (k - 1)!$ and $(k!)^2$.\n\nTherefore, I can divide both sides by $(k - 2)!$ to
    get \\[(k - 1)! 100 = k! (k - 1)!\\]\n\nSimplifying further, I get \\[100 = k
    !.\\]\n\nThis means that $k$ is the smallest integer such that its factorial is
     greater than or equal to 100.\n\nI can try some values of $k$ to find the
    right one.\n\nIf $k = 3$, then $k! = 3! = 6 < 100$.\n\nIf $k = 4$, then $k! =
    4! = 24 < 100$.\n\nIf $k = 5$, then $k! = 5! = 120 > 100$.\n\nTherefore, $k = 5
    $ is the smallest integer that satisfies the equation.
Final Answer: The final answer is $5$. I hope it is correct.

Problem:
What is the largest perfect square factor of 1512?

Solution:
I know that 1512 is divisible by 2, so I can try to factor out powers of 2 until I
    get an odd number.\n\nDividing by 2 once, I get 756.\n\nDividing by 2 again, I
    get 378.\n\nDividing by 2 one more time, I get 189.\n\nNow 189 is odd, so I can
    't divide by 2 anymore.\n\nI notice that 189 is divisible by 3, so I can try to
     factor out powers of 3.\n\nDividing by 3 once, I get 63.\n\nDividing by 3
    again, I get 21.\n\nDividing by 3 one more time, I get 7.\n\nNow 7 is prime, so
     I can't factor it anymore.\n\nI have factored 1512 as 2^3 * 3^3 * 7.\n\nTo
    find the largest perfect square factor, I need to use the highest even powers
    of each prime factor.\n\nThe highest even power of 2 that I have is 2^2.\n\nThe
     highest even power of 3 that I have is 3^2.\n\nThe highest even power of 7
    that I have is 7^0, which is just 1.\n\nSo the largest perfect square factor is
     2^2 * 3^2 * 7^0, which is 36.
Final Answer: The final answer is $36$. I hope it is correct.

Problem:
If $3p+4q=8$ and $4p+3q=13$, what is $q$ equal to?

Solution:

---

I notice that the coefficients of $p$ and $q$ are reversed in the two equations, so I wonder if I can use elimination to get rid of one of them.\n\nI try multiplying the first equation by 4 and the second equation by 3, and then subtracting them to eliminate $p$.\n\nI get $12p+16q-12p-9q=32-39$, which simplifies to $7q=-7$.\n\nDividing both sides by 7, I get $q=-1$.
Final Answer: The final answer is $-1$. I hope it is correct.

**4-shot example for MetaMath Full ICL:**

```
Problem:
In right triangle $PQR$, we have $\angle Q = \angle R$ and $PR = 6\sqrt{2}$.
    What is the area of $\triangle PQR$?

Solution:
Since $\angle Q = \angle R$, we have $\triangle PQR$ is an isosceles right
    triangle.\nTherefore, $PQ = PR = 6\sqrt{2}$.\nThe area of $\triangle PQR$ is
    $\frac{1}{2} \cdot PQ \cdot PR = \frac{1}{2} \cdot 6\sqrt{2} \cdot 6\
    sqrt{2} = \boxed{36}$.
Final Answer: The final answer is $36$. I hope it is correct.

Problem:
The length of the longer side of rectangle $R$ is $10$ percent more than the length
    of a side of square $S.$ The length of the shorter side of rectangle $R$ is
    $10$ percent less than the length of a side of square $S.$ What is the ratio of
    the area of rectangle $R$ to the area of square $S?$ Express your answer as a
    common fraction.

Solution:
Let the side length of square $S$ be $x$.\nThen the longer side of rectangle $R$ is
    $1.1x$ and the shorter side is $0.9x$.\nThe area of square $S$ is $x^2$ and
    the area of rectangle $R$ is $(1.1x)(0.9x) = 0.99x^2$.\nTherefore, the ratio of
    the area of rectangle $R$ to the area of square $S$ is $\frac{0.99x^2}{x^2} =
    \boxed{\frac{99}{100}}$.
Final Answer: The final answer is $\frac{99}{100}$. I hope it is correct.

Problem:
Compute the integer $k > 2$ for which\n\[\log_{10} (k - 2)! + \log_{10} (k - 1)!
    + 2 = 2 \log_{10} k!.\]

Solution:
Using the properties of logarithms, we can rewrite the equation as\n\[\log_{10}
    ((k-2)! \cdot (k-1)!) + 2 = \log_{10} (k!)^2.\]\nThen, using the fact that $
    \log_{a} b + \log_{a} c = \log_{a} (bc)$, we have\n\[\log_{10} ((k-2)! \\
    cdot (k-1)! \cdot 100) = \log_{10} (k!)^2.\]\nFinally, using the fact that $
    \log_{a} b = \log_{a} c$ if and only if $b = c$, we have\n\[(k-2)! \cdot (k
    -1)! \cdot 100 = (k!)^2.\]\nSimplifying, we have\n\[(k-2)! \cdot (k-1)! \\
    cdot 100 = (k \cdot (k-1) \cdot (k-2))!.\]\nCancelling out the common
    factors, we have\n\[100 = k \cdot (k-1) \cdot (k-2).\]\nFactoring, we find
    that $100 = 5 \cdot 5 \cdot 4 = 2^2 \cdot 5^2$.\nSince $k > 2$, we see that
    $k = \boxed{5}$.
Final Answer: The final answer is $5$. I hope it is correct.

Problem:
If $3p+4q=8$ and $4p+3q=13$, what is $q$ equal to?

Solution:
We can solve this system of equations using the method of elimination.\nMultiplying
    the first equation by 3 and the second equation by 4 gives us:\n$9p+12q=24$\
    n$16p+12q=52$\nWe can subtract the first equation from the second equation to
    eliminate the q term:\n$16p+12q-(9p+12q)=52-24$\n$7p=28$\nDividing both sides
    by 7 gives us:\n$p=4$\nSubstituting this value of p into either of the original
    equations gives us:\n$4(4)+3q=13$\n$16+3q=13$\n$3q=-3$\n$q=\boxed{-1}$
Final Answer: The final answer is $-1$. I hope it is correct.
```

**4-shot example for MetaMath Hard ICL:**

```
Problem:
In right triangle $PQR$, we have $\\angle Q = \\angle R$ and $PR = 6\\sqrt{2}$.
    What is the area of $\\triangle PQR$?

Solution:
Since $\\angle Q = \\angle R$, we have $\\triangle PQR$ is an isosceles right
    triangle.\nTherefore, $PQ = PR = 6\\sqrt{2}$.\nThe area of $\\triangle PQR$ is
    $\\frac{1}{2} \\cdot PQ \\cdot PR = \\frac{1}{2} \\cdot 6\\sqrt{2} \\cdot 6\\
    sqrt{2} = \\boxed{36}$.
Final Answer: The final answer is $36$. I hope it is correct.

Problem:
The length of the longer side of rectangle $R$ is $10$ percent more than the length
    of a side of square $S.$ The length of the shorter side of rectangle $R$ is
    $10$ percent less than the length of a side of square $S.$ What is the ratio of
     the area of rectangle $R$ to the area of square $S?$ Express your answer as a
     common fraction.

Solution:
Let the side length of square $S$ be $x$.\nThen the longer side of rectangle $R$ is
     $1.1x$ and the shorter side is $0.9x$.\nThe area of square $S$ is $x^2$ and
     the area of rectangle $R$ is $(1.1x)(0.9x) = 0.99x^2$.\nTherefore, the ratio of
      the area of rectangle $R$ to the area of square $S$ is $\\frac{0.99x^2}{x^2} =
      \\boxed{\\frac{99}{100}}$.
Final Answer: The final answer is $\\frac{99}{100}$. I hope it is correct.

Problem:
Suppose that $y^3$ varies inversely with $\\sqrt[3]{z}$. If $y=2$ when $z=1$, find
    the value of $z$ when $y=4$. Express your answer in simplest fractional form.

Solution:
Since $y^3$ varies inversely with $\\sqrt[3]{z}$, we can write the equation as $y
    ^3\\sqrt[3]{z}=k$, where $k$ is a constant.\nWe are given that $y=2$ when $z=1$
    , so we can substitute these values into the equation to solve for $k$:\n$2^3\\
    sqrt[3]{1}=k$\n$8\\cdot 1=k$\n$k=8$\nNow we can use this value of $k$ to find
    $z$ when $y=4$:\n$4^3\\sqrt[3]{z}=8$\n$64\\sqrt[3]{z}=8$\n$\\sqrt[3]{z}=\\frac
    {8}{64}=\\frac{1}{8}$\n$z=\\left(\\frac{1}{8}\\right)^3=\\frac{1}{512}$\nSo
    when $y=4$, $z=\\boxed{\\frac{1}{512}}$.
Final Answer: The final answer is $\\frac{1}{512}$. I hope it is correct.

Problem:
Let $d$ be a positive number such that when $109$ is divided by $d$, the remainder
    is $4.$ Compute the sum of all possible two-digit values of $d$.

Solution:
If $109$ is divided by $d$ and the remainder is $4$, then $d$ divides $109-4=105$.\
    nThe prime factorization of $105$ is $3\\cdot5\\cdot7$.\nSince $d$ must be a
    factor of $105$, the possible values of $d$ are $1, 3, 5, 7, 15, 21, 35, 105$.\
    nOut of these, only the two-digit values are $15, 21, 35$.\nThe sum of these
    values is $15+21+35=\\boxed{71}$.
Final Answer: The final answer is $71$. I hope it is correct.
```

**4-shot example for MetaMath Easy-to-Hard ICL:**

```
Problem:
How many 4-letter words with at least one consonant can be constructed from the
    letters $A$, $B$, $C$, $D$, and $E$? (Note that $B$, $C$, and $D$ are
    consonants, any word is valid, not just English language words, and letters may
    be used more than once.)

Solution:
There are a total of $5^4 = 625$ possible 4-letter words that can be constructed
    from the given letters.\nTo count the number of words with no consonants, we
    can use the principle of complementary counting.\nSince there are 2 vowels ($A$
    and $E$) and 3 consonants ($B$, $C$, and $D$), there are $2^4 = 16$ words with
    only vowels.\nTherefore, the number of words with at least one consonant is
    $625 - 16 = \\boxed{609}$.
Final Answer: The final answer is $609$. I hope it is correct.

Problem:
Compute the integer $k > 2$ for which\n\\[\\log_{10} (k - 2)! + \\log_{10} (k - 1)!
    + 2 = 2 \\log_{10} k!.\\]

Solution:
Using the properties of logarithms, we can rewrite the equation as\n\\[\\log_{10}
    ((k-2)! \\cdot (k-1)!) + 2 = \\log_{10} (k!)^2.\\]\nThen, using the fact that $
    \\log_{a} b + \\log_{a} c = \\log_{a} (bc)$, we have\n\\[\\log_{10} ((k-2)! \\
    cdot (k-1)! \\cdot 100) = \\log_{10} (k!)^2.\\]\nFinally, using the fact that $
    \\log_{a} b = \\log_{a} c$ if and only if $b = c$, we have\n\\[(k-2)! \\cdot (k
    -1)! \\cdot 100 = (k!)^2.\\]\nSimplifying, we have\n\\[(k-2)! \\cdot (k-1)! \\
    cdot 100 = (k \\cdot (k-1) \\cdot (k-2))!.\\]\nCancelling out the common
    factors, we have\n\\[100 = k \\cdot (k-1) \\cdot (k-2).\\]\nFactoring, we find
    that $100 = 5 \\cdot 5 \\cdot 4 = 2^2 \\cdot 5^2$.\nSince $k > 2$, we see that
    $k = \\boxed{5}$.
Final Answer: The final answer is $5$. I hope it is correct.

Problem:
If $n$ is $1$ less than a multiple of $50$, then what is the remainder when $n^2+2n
    +3$ is divided by $50$?

Solution:
Since $n$ is $1$ less than a multiple of $50$, we can write $n=50k-1$ for some
    integer $k$.\nThen \\[n^2+2n+3=(50k-1)^2+2(50k-1)+3=2500k^2-96k+2,\\] so the
    remainder when $n^2+2n+3$ is divided by $50$ is $\\boxed{2}$.
Final Answer: The final answer is $2$. I hope it is correct.

Problem:
If $3p+4q=8$ and $4p+3q=13$, what is $q$ equal to?

Solution:
We can solve this system of equations using the method of elimination.\nMultiplying
    the first equation by 3 and the second equation by 4 gives us:\n$9p+12q=24$\
    n$16p+12q=52$\nWe can subtract the first equation from the second equation to
    eliminate the q term:\n$16p+12q-(9p+12q)=52-24$\n$7p=28$\nDividing both sides
    by 7 gives us:\n$p=4$\nSubstituting this value of p into either of the original
    equations gives us:\n$4(4)+3q=13$\n$16+3q=13$\n$3q=-3$\n$q=\\boxed{-1}$
Final Answer: The final answer is $-1$. I hope it is correct.
```

