# OpenReview forum: "Easy-to-Hard Generalization: Scalable Alignment Beyond Human Supervision"
_NeurIPS.cc/2024/Conference — NeurIPS 2024 poster_

### Official Review · Reviewer_bhN7 · 2024-06-20

**Soundness:** 3
**Presentation:** 3
**Contribution:** 3
**Rating:** 7
**Confidence:** 4

**Summary:**

This paper attempts to address a forward-looking and challenging question: "Can we limit human supervision to easier tasks, yet enable the model to excel in harder tasks?" (referred to as Easy-to-Hard Generalization). Based on the observation that "evaluation is easier than generation", the authors propose to train a verifier on easy tasks, then make use of its generalization ability to supervise the generator on hard tasks. To harness the complementary strengths of outcome reward models (ORMs) and process reward models (PRMs), the authors introduce the Outcome & Process Reward Model (OPRM), so as to better utilize  Easy-to-Hard evaluators. Through extensive experiments, the authors verify that easier-level evaluators maintain their effectiveness on harder tasks. Further experiments explore the use of the easy-to-hard evaluator as a reward model in reinforcement learning and underscore the potential of using easy-to-hard evaluation to improve easy-to-hard generators.

**Strengths:**

* The problem this paper aims to tackle is promising and challenging.
* The proposed approach is intuitive and has strong motivation.
* This paper is well-written and presents clear ideas.
* Through extensive experiment,  the authors validate that the proposed method of "training a verifier on simple tasks, then leveraging its generalization capability to guide the generator on complex tasks" is workable.

**Weaknesses:**

* **The definition of difficult problems could be further refined.** The scenario considered in this paper is how to enhance the model's ability to perform difficult reasoning tasks when humans cannot provide effective supervisory signals. Given that the model's capabilities vary across the seven subsets in the MATH dataset on which the experiments are based, the definition of difficult problems may be biased. For instance, the LLM performs significantly better on the Algebra subset than on Geometry and Number Theory. Therefore, the improvement at levels 4-5 may primarily result from the performance enhancement in Algebra as it inherently has some evaluation ability for Algebra level 4-5 problems). Thus, it would be better to display the performance of the proposed method in different subsets of MATH, as well as the performance at levels 4 and 5 under different subsets, to prove that it can help the model solve "truly difficult" problems (such as Number Theory level 5).

**Questions:**

See weaknesses. I will consider raising my score if the authors can address my concerns.

**Limitations:**

I think the authors have addressed their limitations.

---

> ### Author Rebuttal · Authors · 2024-08-07
>
> Thank you for your insightful review and positive feedback. We're glad to hear that you appreciate the easy-to-hard generalization problem we’re going on. Your recognition of our proposal and motivations is encouraging. We address your questions below.
>
>
> **Weaknesses**
>
> **W1 (a). The definition of difficult problems could be further refined**
>
> The goal of this paper is not to provide a specific way to split data into easy and hard portions for any arbitrary domain **but to show how we can enable generalization on hard tasks by only supervising the model on easy tasks**. Specifically, in the settings for scalable oversight (aligning superhuman AI), we can treat **all the tasks that humans can annotate as easy, and all the tasks that humans cannot supervise as hard**. This makes a clear definition of “easy” vs “hard” for real-world tasks. It is only for research purposes that the experimental datasets used in the paper, as the reviewer noticed, have a clear division of the difficulty levels. This division helps verify the idea of easy-to-hard generalization (Lines 30-34), where the model is trained only on **easy data (simulating the tasks that humans can label)**, and then generalizing to **hard data (simulating the tasks that humans cannot handle)**.
>
> **W1 (b). For instance, the LLM performs significantly better on the Algebra subset than on Geometry and Number Theory. Therefore, the improvement at levels 4-5 may primarily result from the performance enhancement in Algebra as it inherently has some evaluation ability for Algebra level 4-5 problems). Thus, it would be better to display the performance of the proposed method in different subsets of MATH, as well as the performance at levels 4 and 5 under different subsets, to prove that it can help the model solve "truly difficult" problems (such as Number Theory level 5).**
>
> Thanks for the suggestions. We indeed have conducted the fine-grained analysis of OPRMs’ re-ranking improvements divided by Level and Math Category in Figure 19 and Figure 20 in Appendix N, where Number Theory and Geometry are somewhat more difficult than Algebra. However, we can see in Figure 20 that **for almost all categories, OPRMs’ re-ranking can bring more than a 4% improvement**.
>
> Besides, we have added Figure 2 (Right) in the uploaded PDF to verify the effect under the level 4-5 subset in number theory and geometry. We can see that in this hard subset, **OPRM can bring nearly a 10% improvement**, further demonstrating the feasibility of the easy-to-hard approach and the superiority of OPRM. We will include more experiments on the hard parts of different category subsets in the revision paper. Many thanks for your suggestions.

---

> > ### Comment · Reviewer_bhN7 · 2024-08-10
> >
> > Thanks for your response, my major concerns have been addressed and I have adjusted my score accordingly.

---

> ### Author Response · Authors · 2024-08-12
>
> Dear reviewer bhN7, we are glad our response has addressed most of your concerns. Thank you for increasing your score!

---

### Official Review · Reviewer_VNqz · 2024-07-09

**Soundness:** 3
**Presentation:** 2
**Contribution:** 2
**Rating:** 5
**Confidence:** 4

**Summary:**

This paper addresses the issue that humans cannot always provide helpful demonstrations or supervision on tasks beyond their expertise. Based on the observation that evaluation is easier than generation, the authors propose "easy-to-hard generalization," training a verifier on easy tasks and leveraging its generalization ability to supervise the generator on hard tasks. Experimenting mainly in math reasoning tasks, they demonstrate that easy-to-hard generalization from evaluators can enable easy-to-hard generalization from generators.

**Strengths:**

The paper introduces the Outcome & Process Reward Model (OPRM), which harnesses the complementary strengths of ORMs and PRMs. Experiments show that OPRM is more efficient. It also conducts a systematic experimental setup, testing various generators and evaluators, as well as optimization algorithms (BoN, Weighted Voting, and RL). It provides numerous experimental analyses in scalable alignment.

**Weaknesses:**

While the paper is well-written and presents a solid analysis, several weaknesses need addressing:

1. What is the difference between "easy-to-hard" and "weak-to-strong"? You state that human supervision is available but not reliable in the weak-to-strong setting, but in the [OpenAI paper](https://cdn.openai.com/papers/weak-to-strong-generalization.pdf) it says, "We find that when we naively finetune strong pretrained models on labels generated by a weak model, they consistently perform better than their weak supervisors, a phenomenon we call weak-to-strong generalization." In that study, they finetune GPT-4 with a GPT-2-level supervisor. Can a GPT-2-level supervisor be seen as a verifier on easy tasks? The novelty should be considered.
2. In Table 1, the Full ICL setting performs worse than the Easy-To-Hard ICL setting. How do you explain this? The intuition is that sampling from both easy and hard exemplars may help solve hard problems more effectively than just demonstrating easy exemplars. Although the quality of PRM800K is lower than MetaMATH, your explanation for why the Full ICL setting is worse than the Easy-To-Hard ICL setting is insufficient (line 198).
3. Can you report the average accuracy of verifying each step in a solution when you evaluate the evaluators? Section 3.5 does not explain why this is not reported.
4. When citing images and sections, consider adding automatic jumps for easier navigation.

**Questions:**

1. What is the difference between "easy-to-hard" and "weak-to-strong"?
2. In Table 1, the Full ICL setting performs worse than the Easy-To-Hard ICL setting. How do you explain this?
3. In Table 1, the Easy-To-Hard SFT setting performs slightly worse than the Full SFT setting, which is expected. Do you think the evaluation of the Easy-To-Hard setting is influenced more by the format of problems or by the true grasp of the principles of solving hard tasks? Are there any better evaluation methods? This is an open question, and I would appreciate your thoughts on it.

**Limitations:**

The paper may lack innovation regarding the concept of "easy-to-hard generalization" and the method to achieve it. Many experimental conclusions appear to merely reproduce or support existing works and lack depth.

---

> ### Author Rebuttal · Authors · 2024-08-07
>
> Thank you for your insightful review and constructive feedback. We appreciate your recognition of the paper’s clarity, the proposed OPRM methods, and the solid experiments analysis. We address your concerns and questions below.
>
> **Weaknesses & Questions**
>
> **W1&Q1. What is the difference between "easy-to-hard" and "weak-to-strong"? & Can a GPT-2-level supervisor be seen as a verifier on easy tasks?**
>
> “Easy-to-Hard” (E2H) studies how the model generalizes when trained by **clean** labels on **easy** tasks, while “Weak-to-Strong” (W2S) studies how the model generalizes when trained by **noisy** labels on **all available** tasks.
>
> We listed a few differences between E2H and W2S as clarification, which we’ll add to the revised version of the paper:
>
> - W2S uses weak teacher’s prediction as the supervision, where no human annotation is used for the strong model. E2H uses human annotations, but they are limited to easy problems.
>
> - W2S studies classification or short-answer prediction problems, while E2H studies generative (or long-CoT reasoning) problems.
>
> - The two models used in W2S are the weak teacher and the strong student, which are models of different sizes but trained on the same task. The two models used in E2H are the generator and the evaluator, which can be of the same size, but trained on different tasks (as policy model or as reward model).
>
> - The research question in the W2S analogy is: Can we have the student model outperforming the teacher model? The research question in the E2H analogy is the following: **Can we produce a system (LLM+evaluator) trained on human annotations on easier tasks only but can perform well on harder tasks for which we do not have any human annotations?**
>
> Finally, both E2H and W2S are analogies of scalable oversight, which studies how we can align superhuman AI models.
>
> > “Can a GPT-2-level supervisor be seen as a verifier on easy tasks?”
>
> We believe this is not a well-defined question. We would like to clarify that:1) W2S only studies classification problems, 2) E2H studies generalization of verifiers on hard tasks, not easy tasks.
>
> >  "You state that human supervision is available but not reliable in the weak-to-strong setting, but in the OpenAI paper it says, ..., they finetune GPT-4 with a GPT-2-level supervisor."
>
> The GPT-2-level supervisor in the W2S paper is used to simulate the noisy human supervision on tasks that are too difficult for humans to reliably evaluate.
>
> **W2&Q2. Explanation for why the Full ICL setting is worse than the Easy-To-Hard ICL setting is insufficient (line 198).**
>
> One of our hypotheses is that ICL is mainly performing format learning, so **exemplars of easy problems might be simpler for the model to understand and follow**, whereas the format of difficult problems may be more challenging for the model to grasp.
> Another hypothesis is that **the level of noise in hard data** is likely higher than in easy data. This is like how humans are more prone to making mistakes when annotating difficult questions (inconsistencies in reasoning solutions can also be considered a form of noise), making it difficult for the model to effectively extract knowledge from hard ICL data.
> There is also research [1] suggesting that knowledge is stored in data in a hardness-invariant way. Therefore, selecting hard data for ICL does not necessarily lead to performance improvement.
>
> **W3. Can you report the average accuracy of verifying each step in a solution when you evaluate the evaluators?**
>
> We conducted additional experiments using the PRM800K-test data, which includes correctness annotations for each step, to test our model's ability to distinguish correct reasoning steps. We randomly selected a portion of PRM800K-test data to balance positive and negative samples. The accuracy of the reasoning steps for the three models is as follows:
> | Reward Model         | Step ACC (%) | Outcome ACC (%) |
> |----------------------|--------------|-------------|
> | ORM-PRM800K-7B       | 64.3         | 71.7      |
> | PRM-PRM800K-7B       | **80.4**         | 63.5      |
> | OPRM-PRM800K-7B      | 79.8        |**74.4**     |
>
> This table demonstrates the effectiveness of our trained PRM, showing that PRM has a significantly greater ability to distinguish steps compared to ORM. Additionally, in Figure 2 (Left) of the uploaded PDF, we present the Step ROC curves of three models, where PRM and OPRM exhibit better step discrimination abilities compared to ORM.
> However, it is important to note that a stronger ability to distinguish steps does not necessarily indicate that the evaluator is more helpful for generation. We then also present the Outcome ROC curves of three models on discriminating the final outcome. We collect data generated on MATH500 test set from our 7B policy model. According to the final outcome and groundtruth, we label each data and select a positive-negative balanced set to plot the Outcome ROC curves, where OPRM exhibits better outcome discrimination abilities compared to ORM and PRM. The above table also shows the effectiveness of OPRM on Outcome discrimination ability.
>
> **W4. Citing images and sections**
>
> Thank you very much for your recommendation. We will correct this issue in the revised version.
>
> **Additional Questions**
>
> **Q3. Do you think the evaluation of the Easy-To-Hard setting is influenced more by the format? Are there any better evaluation methods?**
>
> In this paper, we have **controlled all data to have a consistent format**. Therefore, the format will not influence the evaluation results and conclusions presented in the article. Our format is the same for all levels. We released all the data, which are ready for the reviewer to check.
> We also believe accuracy with (greedy, majority voting, best-of-n, and weighted voting) are comprehensive enough to provide an evaluation of the mathematical reasoning tasks.
>
>
> [1] The Unreasonable Effectiveness of Easy Training Data for Hard Tasks, arXiv:2401.0675.

---

> > ### Comment · Reviewer_VNqz · 2024-08-09
> >
> > Thanks for your response. I have read the author's response, and I raise my rating.

---

> > > ### Author Response · Authors · 2024-08-09
> > >
> > > Thank you for raising the score! Please let us know if there are any remaining questions or concerns that we can address!

---

### Official Review · Reviewer_dzZx · 2024-07-11

**Soundness:** 3
**Presentation:** 3
**Contribution:** 4
**Rating:** 7
**Confidence:** 4

**Summary:**

In this paper, the authors propose *easy-to-hard* generalization, which is to train a reward model on simpler tasks and then use it to evaluate the solutions for more difficult tasks. They have conducted in-depth studies on MATH, and also demonstrated effectiveness on the coding benchmark APPS. This work serves as a nice proof-of-concept study of "evaluation is easier than generation", and suggests a new way toward scalable alignment without human supervision.

**Strengths:**

- The idea of *easy-to-hard* generalization is inspiring, and it's nice to see that the idea works out on the challenging MATH dataset.
- The authors have conducted sufficient experiments and detailed analysis, which makes the paper quite worthy to be read and referred to.

**Weaknesses:**

- The definition of "easy" v.s. "hard" is not very clear, and it seems that the proof-of-concept experiments rely on the structure of the benchmarks, as the MATH dataset has 5 divisions of difficulty and APPS has 3. However, when there are not explicit tags of difficulty in a benchmark, what is the authors' definition for "easy" and "hard" in terms of each data sample?

- Following the above weakness, a natural question for the authors is to demonstrate the practical value of this work when compared with recent work that scales synthetic data for MATH (e.g., [1]). Can the massive synthetic data be viewed as a mixture of easy and hard problems + the corresponding solutions? If we treat such mixed data as the easy part and train RMs on it, can we expect similar easy-to-hard generalization? Which kinds of ability would be "unlocked"?

[1] Improve Mathematical Reasoning in Language Models by Automated Process Supervision. https://arxiv.org/abs/2406.06592

- According to the figures shown in the paper, the "Weighted Voting w/ RM" method always  yields better performance as N increases. By comparison, "Best-of-N w/ RM" and "Majority Voting" can plateau or even become worse when N increases from 512 to 1024. Does the weighted voting with RM guarantee the increasing performance, or is it just by accident? Would be great if there are formal explanations to this.

- While scaling the sampling times N has seen improvements, are there certain problems whose correct solutions are never sampled by the LLM when letting N be very large?

- More case studies would be beneficial to provide a more intuitive understanding of how the RM on easy problems generalize to hard ones.

- The authors have conducted in-depth analysis about the comparisons between PRM, ORM, and OPRM on MATH. While I appreciate the experiments, I wonder what the effect of PRM/ORM w.r.t. easy-to-hard generalization is. For example, is it true that we should always adopt PRM when it is possible to get process-based supervision (Lines 219~221)? If this is true, would the results on APPS be better when we could access process-based supervision for code (for example, treat the interleaved comments in a code snippet as that)?

**Questions:**

The idea of "evaluation is easier than generation" is appealing, and it seems that the idea draws inspiration from the assumption that P < NP. However, for some tasks it seems that evaluation shares a similar level of difficulty with generation.

For example, let there be N integers: A_1, A_2, ..., A_N, and the task is to multiply them altogether: A_1 * A_2 * ... * A_N = ? . While doing multiplication is no easy task for LLMs, it seems that evaluating whether some potential answer is the result of A_1 * A_2 * ... * A_N is as difficult as generating the answer of the multiplication, since in either way, one should do it a serial manner (for i in 1,2..N).

Any thoughts/evidence on this?

And broadly speaking, are there cases when evaluation is as difficult as generation, or evaluation is even harder than generation? It would be great if the authors could shed light on this.

**Limitations:**

The authors have adequately addressed the limitations.

---

> ### Author Rebuttal · Authors · 2024-08-07
>
> Thanks for your constructive feedback on our paper. We are glad that you appreciate the inspiring easy-to-hard generalization problem we’re working on, and thank you for acknowledging the thoroughness of our experiments. We address your questions below.
>
> **Weaknesses**
>
> **W1. The definition of "easy" vs "hard".**
>
> The goal of this paper is not to provide a specific way to split data into easy and hard portions for any arbitrary domain but to show **how we can enable generalization on hard tasks by only supervising the model on easy tasks**.  Specifically, in the settings for scalable oversight (aligning superhuman AI), we can treat all the tasks that humans can annotate as easy, and **all the tasks that humans cannot supervise as hard**. This makes a clear definition of “easy” vs “hard” for real-world tasks.
> It is only for research purposes that the experimental datasets used in the paper, as the reviewer noticed, have a clear division of the difficulty levels. This division helps verify the idea of easy-to-hard generalization (Lines 30-34), where the model is trained only on **easy data (simulating the tasks that humans can label)**, and then generalizing to **hard data (simulating the tasks that humans cannot handle)**.
>
> **W2. Can the massive synthetic data[1] be viewed as a mixture of easy and hard problems?**
>
> The easy-to-hard generalization framework is also applicable to data generation methods such as [1]: we can treat all generated question-solution pairs that have been verified by ground-truth as easy problems, while those not verified by ground-truth (or open questions) are considered as hard problems. This is because the easy-to-hard generalization framework does not need to know the ground-truth solutions for the hard problems.
> We leave combining our framework and other methods as future work.
>
> **W3. Does the weighted voting with RM guarantee the increasing performance?**
>
> In our experiments with 7b-34b models, we found that weighted voting with RM is always better than BoN or majority voting. Here’re our insights:
>
> - why weighted voting is better than majority voting? Theorems 1 & 2 in [2] show the convergence of the accuracy with an increasing number of samples. Specifically, the limit is determined by the likelihood of generating the correct answers through all possible reasoning paths (and the likelihood should be viewed as a weighted sum for Weighted Majority Voting). As long as the reward model is “better than random (informally)”, i.e., assigning higher rewards to correct solutions on average, the accuracy limit of Weighted Majority Voting is higher than that of Majority Voting.
>
> - why weighted voting is better than best-of-n? [3] shows that the scaling curve of BoN is
> $R_{bon}(d) = d(\alpha - \beta \cdot d)$, where $d$ is the square root of the KL divergence of the policy and $d = \sqrt{log N - \frac{N - 1}{N}}$.
> This means the performance of BoN will ultimately become worse when reward over-optimization happens (i.e., $d > \frac{\alpha}{2\beta}$).
>
> **W4. Are there certain problems whose correct solutions are never sampled?**
>
> We conducted additional experiments on Pass@N and reported the results in the uploaded PDF. We found there are still some problems that cannot sample a correct answer. More specifically, **Pass@N is highly correlated with difficulty**. As illustrated in Figure 1 in the uploaded PDF, with a larger number of samples, the Pass@N for Level 1 problems is nearly saturated. However, for Level 5 problems, there are still many instances where a correct solution is not sampled.
>
> **W5. More case studies.**
>
> We have included more case studies in Figures 3, 4 of the uploaded PDF.
> Evaluator can help generalize to harder ones in the following ways:
> - The evaluator can **help identify and reduce the confidence of hallucinations caused by misleading information in problems**. As demonstrated in Case Study 1, the solution selected by majority voting with an answer of 36 is misled by the different units of measurement in the problem (2.5 hours and 90 seconds), resulting in an incorrect solution. Then, the ORPM model successfully gives this solution a low score.
> - The evaluator can **assist in reducing the confidence of solutions that misuse mathematical theorems**. In Case Study 2, the majority solution incorrectly applies the theorem "the sum of the exterior angles of a polygon is 360°", leading to erroneous reasoning, and low confidence by the ORPM model.
>
>
> **W6. what the effect of PRM/ORM w.r.t. easy-to-hard generalization is. and would the results on APPS be better with accessing process supervision?**
>
> We compared PRM, ORM, and OPRM in Appendix G, where we found PRMs and ORMs perform similarly, with PRMs slightly outperforming ORMs on hard tasks. However the OPRMs that are trained on the mixed data of PRMs and ORMs significantly outperformed both of them. Hence, we believe we should adopt PRMs and ORMs together for the OPRM, which can complement ORM and PRM.
>
> We believe the results on APPS would be better if we could obtain a human-annotated (or synthetic) PRM dataset for code. However, that’s out of the scope of our paper.
>
> **Questions**
>
> **Q1. Are there cases when evaluation is even harder than generation?**
>
> Not all evaluations are easier than generation. As shown in [4], LLMs might generate content that exceeds their own understanding based on the given context. An LLM might create a **highly coherent and contextually linked story**, but when questioned about the **logical connections** within the story, it may fail to make accurate judgments.
>
> [1] Improve Mathematical Reasoning in Language Models by Automated Process Supervision, arXiv:2406.06592.
>
> [2] An Empirical Analysis of Compute-Optimal Inference for Problem-Solving with Language Models, arXiv:2408.00724.
>
> [3] Scaling laws for reward model overoptimization, ICML 2023.
>
> [4] The Generative AI Paradox: “What It Can Create, It May Not Understand”, ICLR 2024.

---

> > ### Comment · Reviewer_dzZx · 2024-08-12
> >
> > Thank you for the response! Since the added analysis and explanations have resolved most of my concerns, I have raised the score.

---

> > > ### Author Response · Authors · 2024-08-13
> > >
> > > Dear reviewer, we are glad our response has resolved most of your concerns. Thank you for raising your score!

---

### Official Review · Reviewer_3nAE · 2024-07-12

**Soundness:** 3
**Presentation:** 2
**Contribution:** 4
**Rating:** 8
**Confidence:** 4

**Summary:**

- This paper studies the question of how a system can be improved when the performance of a system has surpassed human performance on a task.
- As a testbed, the paper uses problems from the MATH dataset which have been sorted into 5 levels by difficulty.
- The authors first train process-supervised reward models on level 1-3 problems on the MATH dataset.
- They then use the reward models learned from easy problems to supervise policies on hard (levels 4-5) problems on MATH.
- They find that the reward models substantially improve the performance of the policy on hard tasks when used as either reward models in RL or as re-ranking models during inference, despite being only trained on easy problems.

**Strengths:**

- The extensive comparison of different training methods (ReST, DPO, PPO) is useful even outside of the context of the research question.
- The methodological decision to compare both re-ranking and RL on hard problems is very sound and makes me more confident in the conclusion of the paper.
- Although there have been several recent papers on easy-to-hard generalization that establish that easy-to-hard generalization is possible, I think the experimental setup here takes a different angle by showing that the *reward models* learned on easy tasks transfer to harder tasks *in multiple ways* and specifically that evaluators generalize better than generators to hard tasks.

**Weaknesses:**

- The differences between the comparison categories in many cases are small, on the order of 1-2 percentage points. I also did not see any error bars or variance estimates (except maybe in Figure 4, though this is unclear). This makes assessment of the scientific validity of the results a bit more challenging.
- The conclusions were demonstrated on only two tasks and both tasks were formal reasoning tasks. Would the conclusions transfer to natural language reasoning tasks?

**Questions:**

- I found nearly all the tables in the paper hard to read / extract information from.
- It isn't clearly indicated (or at least I couldn't tell) how many times each model was trained, were there multiple runs, etc. I see what looks like error bars on some of the plots, but no explanation of these is given.

---

> ### Author Rebuttal · Authors · 2024-08-07
>
> Thank you for your insightful review and the positive feedback on our paper. We are pleased that you found our proposals interesting and our experiments thorough. It's particularly encouraging to hear that you recognize the novelty of our easy-to-hard generalization approach and our results useful even outside of the context of the research question. We address your questions below.
>
> **Weaknesses**
>
> **W1. The differences between the comparison categories in many cases are small, on the order of 1-2 percentage points. I also did not see any error bars or variance estimates (except maybe in Figure 4, though this is unclear).**
>
> We are only reporting baseline results in Table 1. The main claim of the paper, that the easy-to-hard generalization of evaluators helps generators, is supported by results in Figure 3, Figure 4, Table 2, and Table 3. The accuracy improvements from the reward model are often significant (e.g., comparing weighted voting to majority voting or comparing RL models to SFT models).
> We agree that observing the variance of the error is important. Figures 3 and 4 represent the curves for different combinations of random sampling trials, where the solid curves for the performance average and the shaded areas for the error ranges (performance variance).
>
> **W2. The conclusions were demonstrated on only two tasks and both tasks were formal reasoning tasks. Would the conclusions transfer to natural language reasoning tasks?**
>
> In our easy-to-hard framework, we haven't specialized any assumptions in MATH or Code related to the problem (easy-to-hard generalization) we're studying, so our method should be transferable to other tasks in principle. We left the verification of our method in other domains as future work.
>
> **Questions**
>
> **Q1. I found nearly all the tables in the paper hard to read / extract information from.**
>
> We have added an explanation about the performance variance for each curve and will include the revised explanation in the revised version of our paper.
>
> **Q2. It isn't clearly indicated (or at least I couldn't tell) how many times each model was trained, were there multiple runs, etc. I see what looks like error bars on some of the plots, but no explanation of these is given.**
>
> For the training times, most of the training runs in the paper were only conducted once due to resource constraints, and also because we've observed that the performance was quite stable in our preliminary studies. For the plots, the error bars analysis is indeed included in each curve plot such as Figure 3, 4. We will add more descriptions to the plot in the paper.
>
> Specifically, for all the problems, we sampled 2048 solutions. Taking N=32 on the x-axis as an example, we randomly select 32 solutions from the 2048 rollout samples, record the consensus score (majority voting, weighted voting, and BoN), and repeat this process 400 times. The solid curve represents the mean accuracy of these 400 sampled combinations of solutions, and the shaded margin of each curve represents the performance variance.

---

> ### Comment · Reviewer_3nAE · 2024-08-13
>
> Authors, thank you for the response. I have no further questions. This paper should be a clear accept.

---

> > ### Author Response · Authors · 2024-08-13
> >
> > Dear reviewer, we greatly appreciate your support for our work. Thank you for maintaining your score!

---

### Author Rebuttal · Authors · 2024-08-07

Dear Reviewers and AC,

Thank you all for your time and effort in reviewing our paper. We are grateful to 3nAE, dzZx, and bhN7 for recognizing **the adequacy and novelty of our experiments and motivations** and acknowledging **the importance of the problem we are exploring, easy-to-hard generalization**. We also thank VNqz and bhN7 for recognizing the intuition behind our proposed OPRM method.

Our contributions are well-recognized and can be summarized as:
- We show the potential of easy-to-hard generalization, where models can be guided to **solve complex problems without direct human supervision on these harder tasks**.

- We demonstrate that the easy-to-hard generalization in **evaluator models can effectively guide the generalization of the policy model on challenging tasks**. This underscores the effectiveness of re-ranking strategies and reinforcement learning in leveraging evaluators to achieve performance gains on challenging tasks.

We have added several figures in the uploaded PDF to aid readers in understanding our paper. These figures will also be included in our revised paper:

- Figure 1: The Pass@N curve shows its high correlation with difficulty.
- Figure 2 (Left): The Step ROC Curve and Outcome ROC Curve.
- Figure 2 (Right): The performance of ORPM on Geometry Level 4-5 Problems and Number Theory Level 4-5 Problems.
- Figures 3 & 4: Case studies demonstrating how the evaluator can assist in solving hard mathematical questions.

We sincerely appreciate all the efforts from the reviewers and ACs put into improving our paper. We have responded to every raised concern and hope our response can address them.

Thanks again for all the effort and time.

Best,

Authors

---

### Decision · Program_Chairs · 2024-09-25

**Decision:**

Accept (poster)

**Comment:**

Summary:
The paper addresses an interesting problem of whether, when training an LLM model, we can limit the human supervision to “easier” tasks yet enable the model to generalize well to “harder” tasks where human supervision is unavailable. This problem setting is motivated by scenarios where humans cannot always provide helpful and additionally informative supervision on tasks beyond their capabilities. Based on the observation that, in general, evaluation is easier than generalization, the authors propose to train a verifier on easy tasks and leverage its generalization ability to supervise the generator on more complex tasks. Exploiting the complementarity of ORMs and PRMs, the authors propose a novel framework called OPRMs for the problem. Extensive experimental validation on two sets of data (math and coding) backs the author’s claims on the efficacy of their proposed approach.

Strengths:
* The impact of the problem addressed in the paper is quite significant.
* The techniques proposed in the paper are quite intuitive and are very well-motivated
* The experimental evaluation is quite thorough, further building confidence in the techniques proposed
* The paper is very well written, well structured, and easy to understand

Weakness:
While the reviewers raised some minor weaknesses associated with the paper, the authors' rebuttal was sufficient to address them, prompting multiple reviewers to raise their scores. Some of the weaknesses raised were:
* Imprecise definition of what is an easy problem and what’s a hard problem
* The efficacy of the ideas proposed is validated on only two datasets. Will these ideas generalize to other tasks?

Overall Recommendation:
Overall, this is a high-quality paper addressing an exciting and impactful problem using a novel technique that is very well-motivated. The paper is well-written and easy to understand. While the reviewers had a few objections to the initial version of the paper, most (if not all) of those objections were appropriately addressed in the rebuttal. The unanimous vote is to accept this paper at NeurIPS 2024.